# Optimal Configuration with Capacity Analysis of PV-Plus-BESS for Behind-the-Meter Application

**Cheng-Yu Peng [1], Cheng-Chien Kuo [1,]\* and Chih-Ta Tsai [2]**

[1]  Department of Electrical Engineering, National Taiwan University of Science and Technology,
    Taipei 106335, Taiwan; d10107104@mail.ntust.edu.tw
[2]  Micro Grid and Energy Storage System Division, Smart Energy Business Unit, Tatung Company,
    Taipei 10453, Taiwan; marco.tsai@tatung.com
\*   Correspondence: cckuo@mail.ntust.edu.tw

**Abstract:** As the cost of photovoltaic (PV) systems and battery energy storage systems (BESS) decreases, PV-plus-BESS applied to behind-the-meter (BTM) market has grown rapidly in recent years. With user time of use rates (TOU) for charging and discharging schedule, it can effectively reduce the electricity expense of users. This research uses the contract capacity of an actual industrial user of 7.5 MW as a research case, and simulates a PV/BESS techno-economic scheme through the HOMER Grid software. Under the condition that the electricity demand is met and the PV power generation is fully used, the aim is to find the most economical PV/BESS capacity allocation and optimal contract capacity scheme. According to the load demand and the electricity price, the analysis shows that the PV system capacity is 8.25 MWp, the BESS capacity is 1.25 MW/3.195 MWh, and the contract capacity can be reduced to 6 MW. The benefits for the economical solution are compared as follows: 20-year project benefit, levelized cost of energy (LCOE), the net present cost (NPC), the internal rate of return (IRR), the return on investment (ROI), discounted payback, total electricity savings, renewable fraction (RF), and the excess electricity fraction. Finally, the sensitivity analysis of the global horizontal irradiation, electricity price, key component cost, and real interest rate will be carried out with the most economical solution by analyzing the impacts and evaluating the economic evaluation indicators. The analysis method of this research can be applied to other utility users to program the economic benefit evaluation of PV/BESS, especially an example for Taiwan's electricity prices at low levels in the world.

**Keywords:** behind-the-meter; photovoltaic system; battery energy storage system; time of use rates; electricity

## 1. Introduction

Over the past decade, renewable energy power experienced deployment growth and is accompanied by decreases in system prices. This investment framework has evaluated the effects of dominative public policies, time of use rates (TOU) pricing, low-cost technological improvements, and identified circumstances to financial attraction to Photovoltaic (PV) plus battery energy storage systems (BESS) in behind-the-meter (BTM) market [1]. To minimize the price per kWh of levelized cost of energy (LCOE), the lithium-ion batteries are applied to forecast the dynamics of cost metric in the context—a possible demonstration is its usefulness as an optimally sized battery charged by a PV system [2]. The cost-effective availability of BTM storage is economically viable to incentivize to increase the size in a larger PV system of the optimally sized battery system. The low-cost PV and BESS systems have the potential influence of fundamental shifts in the power sector structure in the United States [3]. For a 3 kW/6 kWh BTM storage system in Australia, the reasonable and practical assumptions would obtain around $80 per year on average, or up to $150 per year in perfect operation over the most recent five years [4].

To meet renewable energy ability and environmental issues, the electric grid is transitioning from central generation to the distributed energy resources (DER) to approach electricity reliability and resiliency targets [5]. However, the electric utility support of PV generation is limited by the reverse condition of power flow at high penetration levels throughout the distribution network, depending on the local infrastructure capacities such as distribution transformer constraints [6]. For instance [7], the specific low voltage networks appeared on unacceptable line congestion and voltage rises under increasing penetration of PV systems in the United Kingdom, and therefore, it is found that BESS may be a reliable method to manage violations exist. For the interactions between the national power system and prosumer households [8], a PV-battery system dispatch model represents that the optimal operation of household batteries minimizes the national electricity supply cost. For the two demand response (DR) programs of Hawaiian Electric Company's revised portfolio approved by the Hawaii Public Utilities Commission [9], the compensation from DR programs is an important value stream to promote the cost-effectiveness of the integrated system of distributed PV and BESS, including fast frequency response and capacity grid service. Simpkins et al. proposed a method for estimating the resiliency during the feasibility stage and optimally sized the system components to minimize the lifecycle cost of electricity, resulting in larger systems and increasing the resiliency to the given site [10]. The suitability of such strategies and act as a guideline is shown to balance the cost optimization process and design robustness of systems [11].

Usually, the investment criterion is considered by capacity allocation and optimal scheme in the green energy business. For the requirement of retail electricity prices above $0.40/kWh and FIT rates below $0.05/kWh, including both Feed-In-Tariff (FIT) and net-energy-metering (NEM) policies [12], PV-plus-BESS systems become better investments than PV-only under three different electricity pricing schemes of US regions from smart meter data of 369 consumers. Utilizing BTM PV-BESS investments meets wider renewable energy goals. The customer relationship with their electricity network resulted that high FIT policy costs in the short-term makes it economically challenging to restrain significant residential PV and BESS adoption in the long-term [13]. Under the new pricing scheme, it is possible to make renewable energy generations, PV and BESS, by using energy management systems and control algorithms [14]. For planning the BTM BESS [8], decreasing the BESS cost leads to rising the penetration rate and reducing the electricity cost with TOU for charge and discharge scheduling according to the evaluation of electricity price, key components cost, and real interest rate. The customer investment in BTM provides the techno-economic scenario analysis to forecast and quantify customer energy transitions [15], and then to select a targeted capacity determines that maximum internal rate of return (IRR) [16]. Optimizing the scheduling of power with BESS achieves the maximum savings compared to purchasing grid power by choosing the BESS capacity with maximum IRR on the PV microgrid [17]. Therefore, the microgrid system can be designed to operate by minimizing the operating cost in BESS flow control according to pricing models [18]. The market integration provides an opportunity to leverage a new scheme of customer energy resources and utility energy resources for the renewable energy transition. The techno-economic analysis of PV and BESS in BTM was applied to an electric vehicle fast-charging station with the small 4-port to obtain a return on investment (ROI) of about $22.4 k over 10 years by 46.5 kW/28.3 kWh BESS [19]. The resiliency analysis explicitly forms the microgrid survivability against a random outage by considering uncertainties associated with PV, the load, and distributed generator failures [20]. An evaluation for microgrid was proposed to get economic benefits from the various grid and BTM services in grid-connecting mode and resilience benefits in islanding mode [21]. The well-designed microgrid with generators PV and BESS held the potential to approach resiliency goals and reduced the net cost of the system during grid operations.

For commercial evaluation using the HOMER Grid software, a rapid cost-effective adoption of PV with BESS is performed as well as a techno-economic ability on the degradation limit and tariff structure. The battery is set only once instead of twice to lower

the net present cost (NPC), and by using 50% lower degrade of initial capacity to a 30% reduction of capital cost during the project lifetime [22]. In addition, the optimal PV and BESS systems at two locations (Arlanda and Karlstad) in Sweden were determined by the indicators of net present value, profitability index (PI), the LCOE, and payback period accordingly [23]. The consideration includes system losses, electricity prices, capital costs, and policy incentives compared between these locations [24]. The synergy between PV and BTM BESS reveals the utility costs reduction in buildings by determining the optimal sizing of the PV and BESS under the assumed utility rate tariffs to result in the largest NPV [25]. For PV plus BESS located in BTM, the impact investigation [26] shows the investment incentives to maximize avoided utility costs and minimize cost-shifting concerns under retail rate structures. The customers accrue economic savings with PV plus BESS, considering a TOU tariff and the steady-state over-voltage disconnection of inverters. The four-quadratic program was compared to lead to the optimization approaches to the designed charge and discharge schedule of residential BESS.

As the cost of a PV system and BESS decreases, the PV system integration BESS applied to the BTM market is growing rapidly. The application of an energy storage system in front-of-the-meter (FTM) [27] can provide services such as bulk energy services, ancillary services, grid support, and renewable energy integration. According to the regulations and demands of the electricity market in various countries, the benefit can be obtained from the services. Under the guidance of energy policies in other countries, the application of PV/BESS in BTM is growing rapidly, and it can mainly provide customer energy management services to utility customers [28]. The incentives include the BESS scheduling with TOU to reduce electricity expense, increase PV self-consumption, and reduce the use of electricity. In addition, the demand response participation can obtain additional revenue, provide emergency power, improve the reliability of power use, etc. It can also bring positive benefits to utilities, such as reducing the load during peak demand, while also increasing the load factor of utilities, reducing the load deviation, and reducing the impact of the intermittent output of renewable energy directly on the grid.

For using the HOMER software to execute the projects, there are many parameters that need to be approximately applied according to their purpose and practical experience with domain knowledge to plan, modify, and operate. Although this paper is not theoretical research, it is a practical site for establishing the systems consisting of PV/BESS through HOMER's operation and planning arrangements. Table 1 lists the hybrid energy systems with various project scenarios. The issue of access to sustainable energy sources is crucial to support the healthcare facilities to deliver services under a grid-connected or an off-grid RHU (rural health units). Based on the experience, the identified load profiles of the equipment can be generally grouped as medical equipment, HVAC (heating, ventilation, and air conditioning), lighting, and office equipment used throughout the day, specifically for the common appliances used in an RHU [29]. To mitigate the dependency on diesel generators where grid extension is not feasible, the generation of electricity may be effectively utilized to the applications of isolated inhabitants through stand-alone PV power systems. Techno-economic analysis is performed to obtain optimal size by empirically assuming typical seasonal load profiles of three distinct island seasons for a single household [30]. To alleviate this challenge of the intermittent nature of renewable energy sources (RESs), it is common practice to integrate RESs by efficient batteries playing the leading role utilized as stationary energy storage systems. The investigations can be concluded that Li-ion batteries require 40% less power as compared to lead-acid batteries, and Li-ion batteries provide lower NPC and COE for photovoltaic grid-connected systems [31]. Due to the low price of diesel fuel in Iran, the competitiveness of such renewable systems with diesel generator systems is low. To increase the economic acceptability, the analysis leads to the finding that Iran requires the government's support and incentive schemes to the competitiveness in energy supply systems [32]. The electrification of consumption is implemented for three different PV residential households with incorporating feed-in Tariff of day (ToD) tariffs regulation/net metering process. The feasibility presents the executable

option and the possible consequence of a social benevolent policy for the dissemination of decentralized grid-connected renewable energy systems [33]. Under the local policies' limits on maximum PV size in Queensland, the maximized PV size is able to maximize the Queensland residents' benefits and meets the requirement of minimizing the total costs of system investment related to electricity consumption during the system's lifetime. The result also obtained the best slope of the PV panel located at 20°~25° dependent on the different cities [34].

**Table 1.** The hybrid energy systems with various project scenarios.

| Hybrid Energy System | Application Scenarios | Method-ology | Performance | Restriction Conditon | Performance Evaluation | Case Study in the Site |
|---|---|---|---|---|---|---|
| • PV/Battery | The sustainable energy source to support the healthcare facilities | HOMER | The healthcare facilities to deliver services under a grid-connected or an off-grid | Different load profiles of equipment | NPC, COE, RF | Philippine rural health units (RHU) [29] |
| • Diesel/PV/Battery/Hydrogen | To mitigate the dependency on diesel generator | HOMER | Optimal the size of stand-alone configuration for the seasonal load profiles of three distinct island seasons for a single household | The island climatic conditions, and the typical seasonal load profiles | NPC, COE | Indian isolated island Andaman and Nicobar island [30] |
| • Battery | To alleviate the intermittent nature of renewable energy sources | HOMER + Matlab | Considering the different stationary application profiles by the analysis of lead-acid and Li-ion batteries | The specifications and application profiles of battery systems | NPC, COE, RF | Euro dollars evaluation [31] |
| • Battery <br> • PV/Battery | Evaluate the competitiveness of such renewable systems | HOMER | The suggestion of an off-grid power supply system for PV-diesel-battery based on the government's support and incentive schemes | The low price of diesel fuel in Iran | NPC, COE, COP, carbon emission | Iran [32] |
| • Diesel/PV/Battery/Hydrogen | The electrification of consumption for the dissemination | HOMER | The better performance of PV system stimulates the investment opportunity | Diesel fuel price fluctuation | NPC, COE, RF | New Delhi, India [33] |
| • PV/Battery | Residential PV system to optimize the size and slope of PV array | HOMER | Maximize the Queensland residents' benefits for 4 typical climate zones | The local policies' limits on maximum PV size | NPC, COE, ROI, carbon emission | Queensland, Australia [34] |

Taiwan promulgated the regulatory measures on energy saving improvement and renewable energy supply during two development stages: the growth period (annually 5.6%) of 1990–2005 and the decoupling period (annually 0.5%) of 2005–2018 to relevant sustainability indicators based on greenhouse gas emissions [35]. Subsequently, Taiwan announced that the policy would replace nuclear power with renewable energy by 2025, which accounts for approximately 4.43% of its total energy supply (or 8.30% of total electricity supply) [36]. Since Taiwan passed the amendment to the "Renewable Energy Development Act" in 2019, the policy tools strongly promote the energy development of the industry and the evaluation tools determine the installation ranking according to criteria

weighting conditions [37]. According to the policy, the electricity consumers above a certain contracted capacity must be stipulated by one of the requirements in the following: (1) install a certain proportion of renewable energy equipment for power generation, (2) install a BESS, (3) purchase a certain proportion of green power of Taiwan Renewable Energy Certificate (T-REC), or (4) pay the fine. The primary specification object is set on the electricity consumer with a contract capacity of 5 MW or more by the implementation rules with the obligation fulfilled within five years. After the implementation, a rolling review every two years will gradually expand the primary specification object. The requirement presents 10% of the contract capacity to the installed capacity of renewable energy equipment for power generation while the BESS's capacity meets 2 h [38].

Based on Taiwan's energy policy, this article mainly discusses the economic benefits of integrating the PV system with BESS and applying BTM. We use the HOMER Grid software to conduct electrical and economic simulation analysis with actual cases. According to the previous economic analysis of PV or PV/BESS works of literature, the traditional methods directly apply the load demand or electricity price evaluations to set PV capacity, but the constraints of the excess electricity fraction are less considered to the correspondence between renewable fraction (RF) and excess power. Usually, the operation of excess power uses power curtailment by the solar inverter and reveals a waste of excess PV power. Therefore, the excess electricity fraction is limited to 3% to a novel precise analysis and effective use of energy for the economic benefit evaluation of PV/BESS. The energy arbitrage revenue can be obtained by calculating the optimal economical capacity allocation of a PV system and BESS in the conditions of meet load demand and PV's RF constraints at the same time. The HOMER software is a simulation analysis tool. The analysis method requires users to plan according to their needs and it can carry out relevant research and analysis according to different research needs.

This study uses HOMER simulation to propose the economic benefit analysis method of PV/BESS applied to BTM, which is to consider the effective use of renewable energy. Especially an analysis method is considered by the restriction of excess electricity, and to filter out the solutions that meet the restriction of excess electricity from the simulation results (excess electricity is set to 3% in this study) to find out the capacity of PV/BESS solutions under the conditions of different proportions of renewable energy. The analysis method to most project benefits is the part that has not been considered in the economic benefit analysis literature in the past, and this method is proposed for the engineer application of renewable energy to the BTM economic benefit analysis. The design plan is under comprehensive considerations for the electricity users to evaluate and choose the system plan.

## 2. Methodology

The analysis of the economic benefits of PV/BESS application of BTM in this study shows that the load is the priority to use the electricity generated by the PV system to reduce the demand for purchased electricity. The BESS with TOU price is considered to adjust the charge and discharge schedules during the period of off-peak electricity price and the period of on-peak electricity price, respectively, in order to obtain the profit from the electricity price difference and reduce the cost of annual contract capacity. Therefore, the sum above two reductions in electricity bills is the PV/BESS project benefit. This research uses the contract capacity of the actual industrial user as a research case and performs PV/BESS techno-economic simulation analysis through the HOMER Grid software.

According to Taiwan's policy of contracted capacity, the objective of the analysis shows the reduced contracted capacity related to the most economical PV/BESS capacity allocation and optimal contract capacity scheme under the condition that the electricity demand is met and the PV power generation is fully used. Figure 1 shows the flowchart of the economic benefit analysis. Without a waste of excess PV power, the excess electricity fraction is limited within 3% to the analysis and effective use of energy for the economic benefit evaluation of PV/BESS.

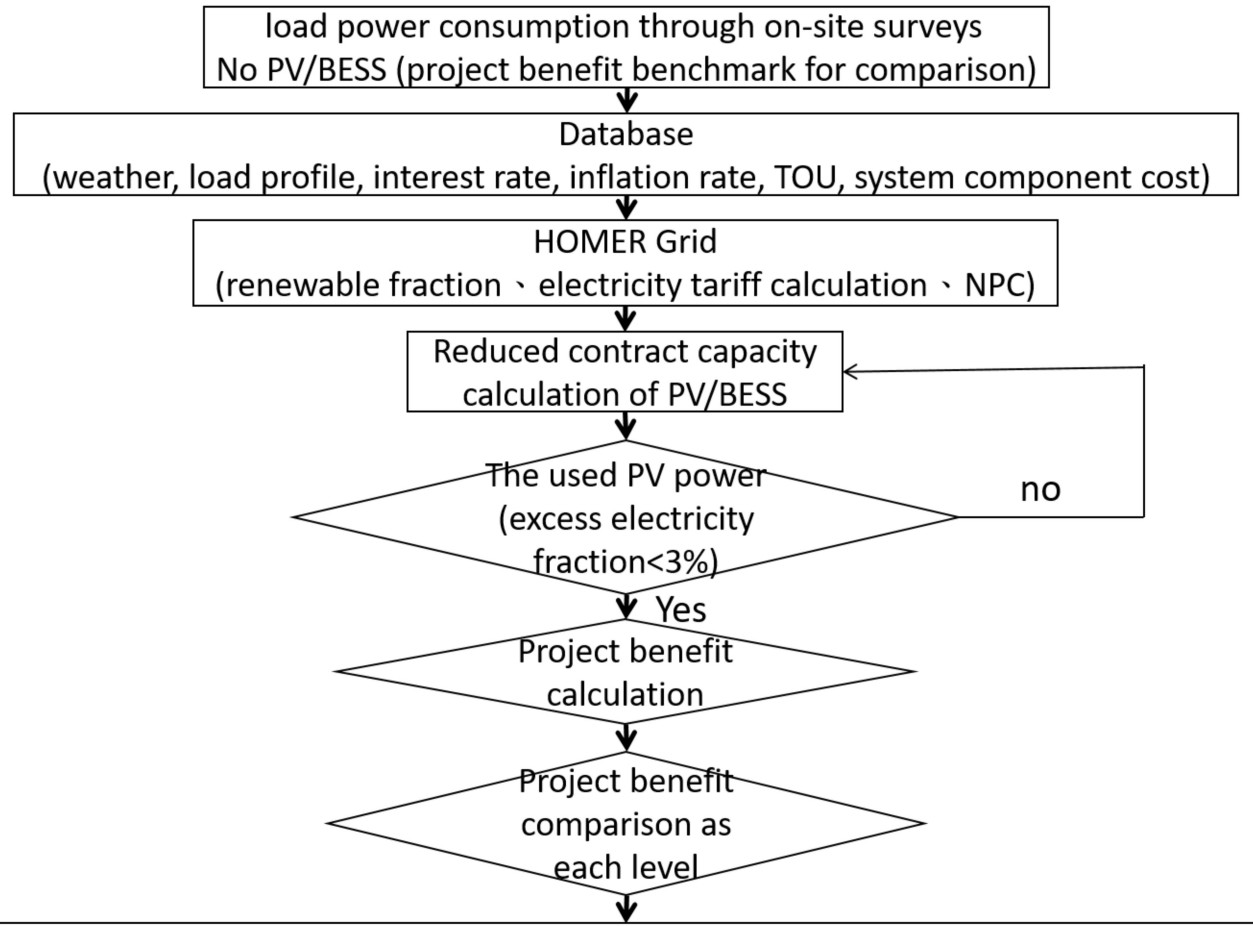

**Figure 1.** The flowchart of the economic benefit analysis.

From several cases of contract capacity reduction, the economical combinations of the PV system and BESS capacity are analyzed to find the most economical combination scheme of contract capacity reduction and the corresponding PV/BESS capacity. The research methods and analysis are as follows:

(1) Through on-site surveys, the load power consumption from industrial users in 2019 is obtained. The basic data for simulation is as statistic interest rate, inflation rate, and time price of electricity in Taiwan in recent years, as well as the cost of equipment obtained from the market survey;

(2) According to the load conditions of industrial users, the result analyzes the electricity expense without the energy storage system installed as a benchmark for comparison of project benefits;

(3) According to the conditions of load and TOU electricity price, the simulation has considered that PV electricity can be fully used and the excess electricity fraction must be limited to less than 3%. The result addressed to calculate the highest project benefit for PV/BESS capacity allocation schemes under different contract capacities, and they are corresponding electrical and economic results, respectively;

(4) According to the simulation result of (3), the most economical scheme is from several reduction cases of contract capacity to find the combination plan of contract capacity with the highest project benefit and their operation capacity of PV/BESS;

(5) Based on the combined plan of contract capacity of the highest project benefit and PV/BESS capacity, the sensitivity analysis carries out to know the degree of impact on economic benefits by the parameters such as global horizontal irradiation (GHI), electricity price, key components cost, and real interest rate.

### 2.1. Simulation Software Description

HOMER Energy LLC is one of the world's leading green energy power system software companies related to distributed generation and micro-grid modeling. The software was originally developed by NREL (National Renewable Energy Laboratory) and acquired by Underwriter Laboratories Inc. in December 2019. This paper studies the applications of the HOMER Grid in the HOMER Energy analysis software series. HOMER Grid reveals engineering analysis and economics information in one combined model to perform rapidly complex calculations with the comparisons of multiple components and design outcomes, especially for technical points of cost-competitive evaluations. It is suitable to consider various options for minimizing project risk and reducing energy expenditures in this project of PV-Plus-BESS of BTM [39].

### 2.2. Weather Conditions

The weather condition relates to the latitude of 25.079° and longitude of 121.189° of the research site in this study into the HOMER Energy software to obtain NASA's global horizontal irradiation data (GHI) and temperature statistics through the internet service. The area shows an annual GHI of $1.478 \times 10^3$ kWh/m$^2$/year and the daily average GHI of 4.05 kWh/m$^2$/day with abundant sunshine and is suitable for the development of PV applications. As shown in Figure 2, the average GHI of each month is from 2.23 kWh/m$^2$/day to 6.84 kWh/m$^2$/day. Figure 3 shows the average temperature of monthly temperature distribution with an annual average temperature of 21.69 °C, ranging from 14.91 °C to 27.81 °C. The highest and lowest temperature months are located in July and January, respectively.

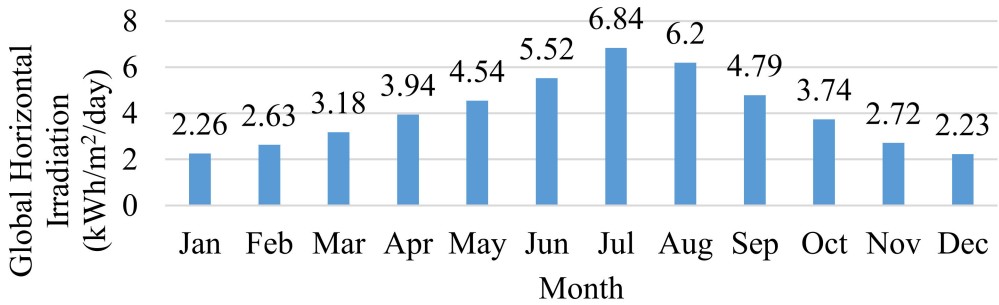

**Figure 2.** Monthly global horizontal irradiation distribution.

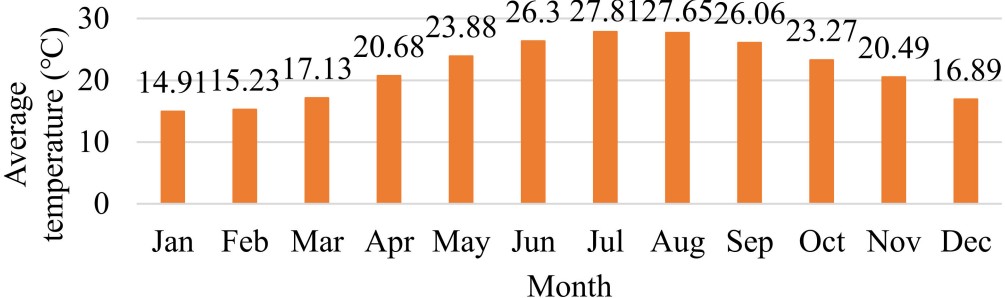

**Figure 3.** Monthly temperature distribution.

### 2.3. Load Consumption Profile

This research case is an industrial user located in northern Taiwan, and the statistical period for the load data sampled every 5 min is from January 2019 to December 2019. The annual electricity consumption is $30.331 \times 10^3$ MWh/year, and the average monthly electricity consumption is $2.527 \times 10^3$ MWh/month. The monthly electricity consumption is differently related to the monthly production volume of the factory as shown in Figure 4, the highest electricity consumption of $2.791 \times 10^3$ MWh/month is in May. However, the lowest electricity consumption is $2.003 \times 10^3$ MWh/month in February, because of the Chinese New Year in Taiwan.

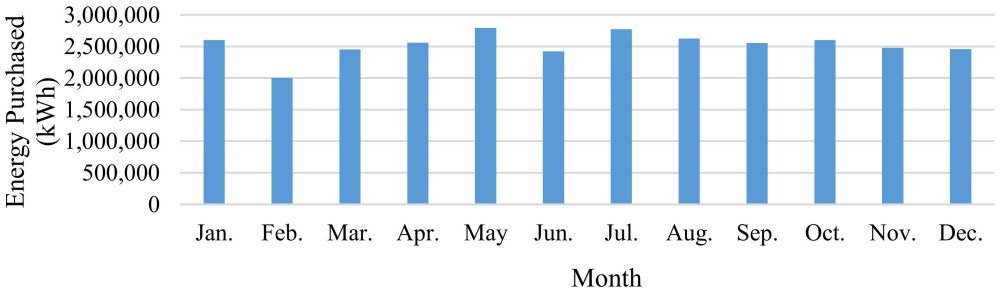

**Figure 4.** Monthly electricity consumption distribution.

The average daily load profile of annual power consumption shows the periods of peak power consumption from 8:00 to 12:00 and from 13:00 to 16:00, as shown in Figure 5, and the daily power consumption of annual average is 83.1 MWh/day. However, because most of the high power consumption is mainly during the daytime, a PV system with daytime power generation characteristics can reduce the electricity expense (energy purchased from utility electricity).

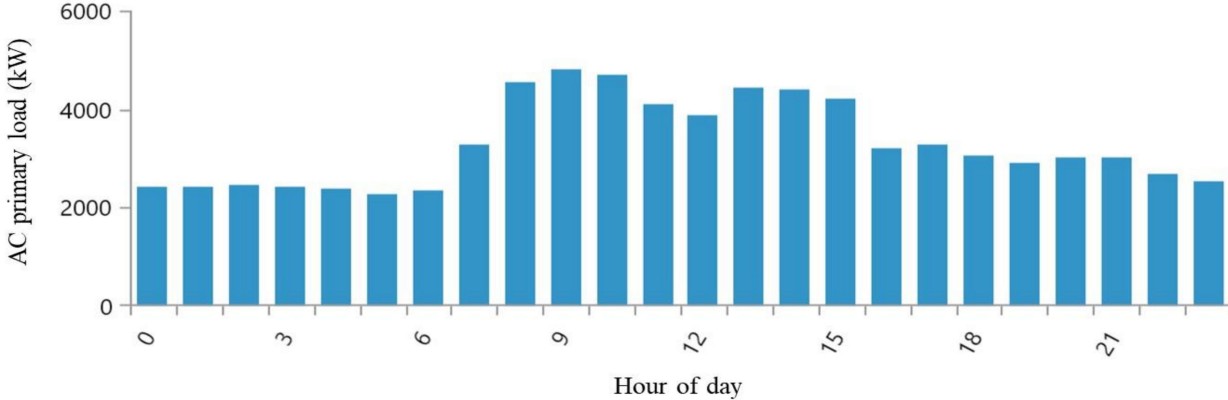

**Figure 5.** The load profile for the average daily power consumption of annual energy.

The load profile shows results based on the power data of users in 2019. The power data imported into the software is raw data, with each recorded every 5 min. The load power in the 24 h in Figure 5 is the average value of 365 days in each period. The peak power of 7378 kW is the highest load power ever experienced in the year. The user schedule of load consumption power shows an average of 3.463 MW. As shown in Figure 6, the frequency distribution of load consumption power can be accounted for by 60.13%, the most in this range between 1.75 MW and 3.5 MW. The frequencies locate the highest at 2.75 MW with accounting for 13.09%, the proportion of 7.11% at 6 MW or more, and the proportion of 0.1% at 7 MW or more. Therefore, if the energy storage system (PV/BESS) can provide a short time of 6 MW or more electricity demand, it can reduce electricity expenses for the contract capacity.

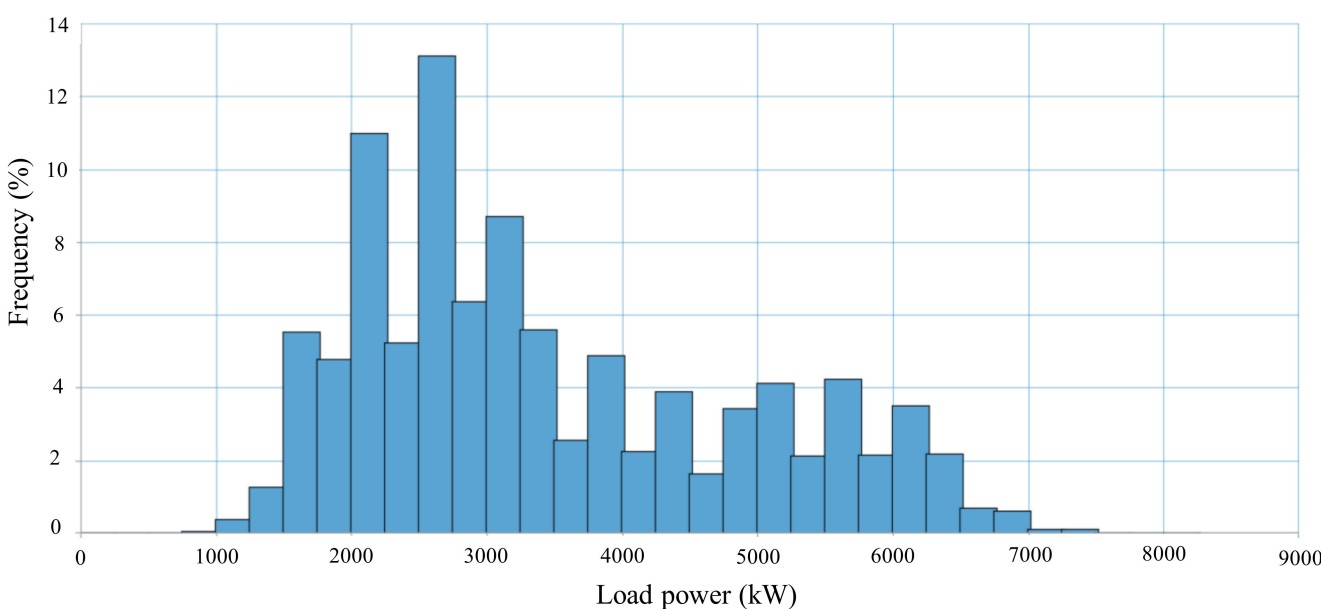

**Figure 6.** The frequency distribution of load consumption power.

### 2.4. Electricity Tariff Calculation

According to the listed electricity price of Taiwan Power Company (TPC), it is mainly divided into five type plans as shown in Table 2. There are the 1st type to the package price of public construction, the 2nd and 3rd types to have TOU options, and the 4th and 5th types based on the voltage level of different electricity consumption. The 4th and 5th types correspond to different contract capacity fees and can choose two-stage or three-stage TOU.

**Table 2.** The electricity tariff of TPC and its application in Taiwan.

| Item | Type | Application |
|:---:|:---:|:---:|
| 1 | Package price [1] | Public street lighting, alarms |
| 2 | Meter rate lighting service | Houses, small shops, offices, institutions, schools, and other institutions |
| 3 | Low voltage (LV) | Power supply below 600 volts for public offices, schools, supermarkets, small shopping malls, small and medium-sized factories |
| 4 | High voltage (HV) | Power supply from 600 volts to 22,800 volts with contract capacity more than 0.1 MW for factories, department stores, public institutions, schools |
| 5 | Extra high voltage (EHV) | Power supply more than 22,800 volts with contracted capacity of 1 MW or more for factories, MRTs, railways, and airports |

[1]: Electricity charges are only according to the number of electric equipment and its rated power consumption, without an electric meter installation.

According to TPC regulations, the basic electricity fee will be charged twice in that month if the average power consumption per 15 min exceeds less than 10% of the contract capacity, and the basic electricity fee will be charged triple if the contract capacity exceeds more than 10%. The electricity price type of this research case is the 5th type related to the electricity voltage level which is an extra high voltage (EHV) of 69 kV. Moreover, the three-stage time of TOU is used, and the contract capacity is 7.5 MW, as shown in Table 3. For the summer month, the contract capacity price of the basic electricity fee is 1.35 times that of the non-summer month. The price of three-stage electricity during the on-peak

period of the summer month is 3.57 times that of the off-peak period, and the price during the mid-peak period of the non-summer month is 2.28 times that of the off-peak period.

**Table 3.** Three-stage electricity price of extra high voltage.

| Basic Charge | Contract Capacity Charge | | | Summer Months (1 June~30 September) | Non-Summer Months (1 June~30 September) |
|---|---|---|---|---|---|
| | | | | 7.1246 | 5.2656 |
| Energy Charge | Monday to Friday | On-peak | Summer months | 10:00~12:00 13:00~17:00 | 0.1511 | - |
| | | Mid-peak | Summer months | 07:30~10:00 12:00~13:00 17:00~22:30 | 0.0941 | - |
| | | | Non-summer months | 07:30~22:30 | - | 0.0911 |
| | | Off-peak | 00:00~07:30 22:30~24:00 | | 0.0423 | 0.0400 |
| | Saturday | Mid-peak | 07:30~22:30 | | 0.0567 | 0.0541 |
| | | Off-peak | 00:00~07:30 22:30~24:00 | | 0.0423 | 0.0400 |
| | Sun. and off-peak day | Off-peak | All day | | 0.0423 | 0.0400 |

Figure 7 shows the electricity expenditure of the research site in each month in 2019. Related to the statistics of utility fees the total annual electricity cost is $2.85 \times 10^6$/year, including the energy charge of $2.321 \times 10^6$/year accounting for 81.42% and the contract capacity charge of $529.488 \times 10^3$/year accounting for 18.58%. The highest and lowest electricity rates locate in July and February (1.78 times between highest and lowest), and there is no penalty for electricity consumption exceeding the contract capacity of its full year.

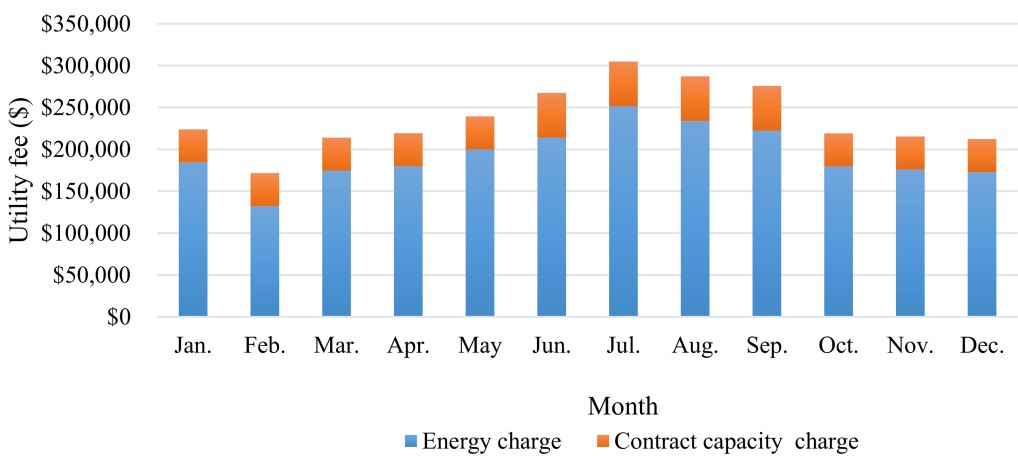

**Figure 7.** Electricity expenditure of each month in 2019.

### 2.5. Economic Assignment Criteria

To have calculations on the cost-effectiveness of different system configurations, the descriptions and formulas of the economic evaluation indicators used in this study are shown in Section 2.5.1, Section 2.5.2, Section 2.5.3, Section 2.5.4, Section 2.5.5, Section 2.5.6, Section 2.5.7, Section 2.5.8 [37], respectively.

#### 2.5.1. Annual Real Interest Rate

The real interest rate is approximately the difference between the nominal interest rate and the inflation rate, which is the rate after allowing for inflation. The annual real discount rate is calculated from the nominal interest rate and expected inflation rate according to

the real interest rate to convert costs between one-time expense and annualized expense. From NPC, the annual real interest rate leads to discount factors and annualized costs by the HOMER Grid software.

$$i = \frac{i' - f}{1 + f},$$ (1)

where:

$i$ = annual real interest rate (%);
$i'$ = nominal interest rate (bank board rate) (%);
$f$ = expected inflation rate (%).

### 2.5.2. Net Present Cost (NPC)

The NPC represents a life-cycle cost of components related to the values of installation and operation over the project lifetime in the HOMER Grid software. Total NPC indicates the summation of t-year cash flow over the annual real interest rate and initial capital cost during the project lifetime. For the t-year cash flow of definition, the expenditure and the income in the HOMER Grid software are positive and negative, respectively. In the above, the expenditure includes capital cost, operation cost, replacement cost, maintenance cost, etc., and the income includes electricity selling and component remaining value after the life cycle.

$$
\begin{aligned}
NPC &= -CF_0 + \left\{ \frac{CF_1}{(1+i)^1} + \frac{CF_2}{(1+i)^2} + \frac{CF_3}{(1+i)^3} + \ldots + \frac{CF_N}{(1+i)^N} \right\} \\
&= -CF_0 + \sum_{t=1}^{N} \frac{CF_t}{(1+i)^t}
\end{aligned}
$$ (2)

where:

$CF_t$ = the cash flow of the t-year ($);
$i$ = the annual real interest rate (%);
$N$ = the project lifetime (year);
$t$ = the number of year (year);
$CF_0$ = the initial capital cost ($).

### 2.5.3. Capital Recovery Factor (CRF)

The capital recovery factor is a rate to calculate the present value between the real discount rate and the number of years, which presents an annuity related to a series of equal annual cash flows during the project lifetime. The Equation (3) for the capital recovery factor is shown in the following:

$$CRF(i, t) = \frac{i(1 + i)^t}{(1 + i)^t - 1}$$ (3)

where:

$t$ = the number of year (year)
$i$ = the annual real interest rate (%)

### 2.5.4. Levelized Cost of Energy (LCOE)

The definition of levelized cost of energy (LCOE) in this study is the average NPC of electricity generation per kWh for load consumption plan over the lifetime. LCOE is the unit of $/kWh calculated by the total annualized cost (TAC) dividing by the total annualized load consumption, where TAC is a product of NPC and CRF. The equation is shown in the following:

$$TAC = |NPC| * CRF(i, N),$$ (4)

$$LCOE = \frac{TAC}{E_{prim}},$$ (5)

where:

TAC = the annualized value of NPC ($/year);
$E_{prim}$ = the total annualized load consumption (kWh/year);
$N$ = the project lifetime (year).

### 2.5.5. Project Benefit (PB)

The project benefit represents the unit of $ to the NPC difference between the current case system and the reference case system, and the reference case system is a scenario without a BESS installed. A positive value of PB indicates saving money in the current case system over the project lifetime compared to the reference case system, whether the current case system with positive PB is preferred as an investment option with the reference case. The equation is shown in the following:

$$PB = (-CF_{c,0} + \sum_{t=1}^{N} \frac{CF_{c,t}}{(1+i)^t}) - (-CF_{ref,0} + \sum_{t=1}^{N} \frac{CF_{ref,t}}{(1+i)^t})$$
$$= NPC_c - NPC_{ref} \tag{6}$$

where:

$NPC_c$ = the NPC of the current case system ($);
$NPC_{ref}$ = the NPC of the reference case system ($).

### 2.5.6. Return on Investment (ROI)

The return on investment represents performance on the yearly cost savings to evaluate the profitability of a number of different investments. The software calculates the amount of return on the investment's cost, and the benefit is divided by the cost of the investment to ROI as shown in the following:

$$ROI = \left( \frac{(-CF_{c,0} + \sum_{t=1}^{N} CF_{c,t}) - (-CF_{ref,0} + \sum_{t=1}^{N} CF_{ref,t})}{N \times \left( C_{cap,c} - C_{cap,ref} \right)} \right) \times 100\%, \tag{7}$$

where:

$C_{cap,c}$ = the capital cost of the current case system ($);
$C_{cap,ref}$ = the capital cost of the reference case system ($).

### 2.5.7. Discounted Payback Period (DPP)

The discounted payback period represents the unit of year to evaluate the time with capital budgeting procedure to determine the profitability by taking the cumulative discounted net cash flow to offset the initial investment. The equation is shown in the following:

$$DPP = t_{full} + \frac{C_{cap,last}}{\frac{CF_{c,t_{full}+1}}{(1+i)^{t_{full}+1}} - \frac{CF_{ref,t_{full}+1}}{(1+i)^{t_{full}+1}}}, \tag{8}$$

where:

$t_{full}$ = the time before the accumulated cash flow (year);
$C_{cap,last}$ = the remaining unrecovered capital cost after accumulated cash flow ($).

### 2.5.8. Internal Rate of Return (IRR)

The IRR is a way to estimate an investment's rate of return to the profitability of potential investments. IRR is the discount rate that sets the net present value of all cash flows from the investment equal to project zero in financial cash flow analysis. In this study, IRR is calculated by using the annual cash flow value without considering the real

interest rate, and it is not relative to real interest rate changes. Therefore, Equation (9) of IRR is applied to calculate the IRR result by a spreadsheet calculator in the program, and the lowest level of IRR is acceptable to justify the investment.

$$0 = (-CF_{c,0} + CF_{ref,0}) + \sum_{t=1}^{N} \frac{CF_{c,t} - CF_{ref,t}}{(1 + IRR)^t}. \tag{9}$$

### 2.6. Electrical Assignment Criteria

The simulation has considered that renewable energy electricity can be applied, and the excess electricity fraction must be limited to less than 3% according to the conditions of load and TOU electricity price. The formulas to limitation are described below to the definition and description of the HOMER Grid [30].

#### 2.6.1. Renewable Fraction (RF)

The RF has important contributions on sizing the grid-connected renewable systems and managing electricity to the electricity cost. RF represents that the energy delivered to load power consumption originated from the fraction of renewable sources. The equation is shown in the following:

$$\text{RF} = \left(1 - \frac{E_{non-ren} + H_{non-ren}}{E_{served} + H_{served}}\right) \times 100\%, \tag{10}$$

where:

$E_{non-ren}$ = the nonrenewable electrical production (kWh/year);
$H_{non-ren}$ = the nonrenewable thermal production (kWh/year);
$E_{served}$ = the total electrical load served (kWh/year);
$H_{served}$ = the total thermal load served (kWh/year).

#### 2.6.2. Excess Electricity Fraction

The excess electricity fraction is the quotient equal to the total excess electricity divided by the total electrical production. The excess electricity represents the minimum output that exceeds the load, so it is unable to serve a load or charge batteries. The equation is shown in the following:

$$\text{Excess electricity fraction} = \frac{E_{excess}}{E_{prod}} \times 100\%, \tag{11}$$

where:

$E_{excess}$ = the total excess electricity (kWh/year);
$E_{prod}$ = the total electrical production (kWh/year).

## 3. Grid-Connected PV/BESS System Description

The system concept operates in Section 3.1 as the following description of each device, including Section 3.2 PV system, Section 3.3 BESS, Section 3.4 power conversion system (PCS), and the system dispatch strategy in Section 3.5.

### 3.1. Grid-Connected PV/BESS System Schematic

The simulation schematic of the Grid-connected PV/BESS system is illustrated in the analysis cases in the study as shown in Figure 8, which is composed of the grid, PV system, storage system, power conversion system (PCS), and load between AC bus and DC bus. In accordance with the load power curve, the PV system using the AC couple type supports directly the AC load electricity during the daytime, in order to reduce the energy conversion loss. Under the condition that the electricity demand is met and the PV

power generation is fully used, the target is to find the most economical PV/BESS capacity allocation and optimal contract capacity scheme.

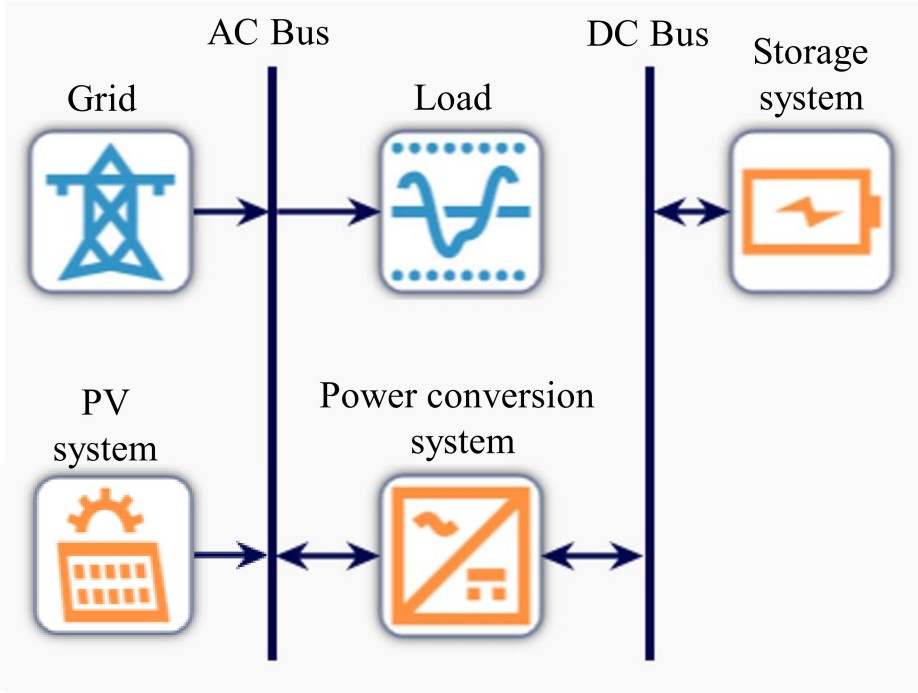

**Figure 8.** Schematic diagram of the grid-connected PV/BESS system.

*3.2. PV System*

The power generation method of the PV module is to convert the energy of sunlight into DC energy of power electricity through the PV effect of a solar cell. The rated capacity of the PV module in this study is the output power of 330 Wp by using module type of monocrystalline silicon solar cell, the conversion efficiency of 19.78%, temperature coefficient of $-0.39\%/°C$ and the lifetime of more than 25 years. The tilt angle of $10°$ in the factory roof is set by the installation angle of the PV module. The PV system is composed of 32 modules connected in series to form a string, and 24 strings are connected in parallel to form a subsystem with a capacity of 253.44 kWp. However, for the convenience of simulation setting and analysis, it is simplified to 0.25 MWp at each level. The parameters are set as the lifetime of 20 years and the PV derating factor of total system loss of 85%. The output power formula in the HOMER Grid software is as shown in Equation (12) [40,41].

$$P_{PV} = P_{pv,STC} f_{pv} \frac{G_T}{G_{T,STC}} \left[1 + K_p (T_C - T_{STC})\right], \tag{12}$$

where:

$P_{pv,STC}$ = PV system rated power (kWp);
$f_{pv}$ = PV derating factor (%);
$G_T$ = the solar irradiance on the surface of the PV module (kW/m$^2$);
$G_{T,STC}$ = the solar irradiance under the standard test conditions (1 kW/m$^2$);
$K_P$ = the temperature coefficient of PV module (%/°C);
$T_C$ = the operation temperature of the PV module (°C);
$T_{STC}$ = the temperature of the PV module under the standard test conditions (25 °C).

### 3.3. Storage System

In this study, the high-energy-density and high-safety lithium ferrous phosphate battery (LFP) was used for economic benefit analysis. The battery cell used in this study has a rated specification of 3.2 V/280 Ah. The upper limit of the current can allow for continuous charging or discharging with 140 A, and the time would require to fully charge or discharge with 2 h, which is a 0.5 C-rate battery. The HOMER simulation software adjusts the current according to the power demand for charging or discharging, and the upper limit is 140 A, which is not a constant value. The battery module is made up of 14 battery cells in series with a rated power of 44.8 V/280 Ah, and the battery rack is made up of 17 battery modules in series with a rate of 761.6 V/280 Ah and a rated capacity of 213.2 kWh. The operating voltage of the battery cabinet is 666.4~856.8 V, the minimum state of charge (SOC) is set to 20%, and the round trip efficiency is set to 92%.

The lifetime presents the number of discharge-charge cycles to meet specific performance criteria of the battery. The operating lifetime throughout of 967.765 MWh of the battery is affected by the parameters (the rate and depth of cycles) and by the environmental conditions (temperature and humidity). The warranty of a commercial battery is in the range of 5–10 years according to industrial experiences. The related standard such as IEC62620 can provide the parameters to lifetime according to the customer's declarations. The battery wear cost is the cost of cycling energy through the storage system calculated by Equation (13), and the storage life is limited by throughput. HOMER assumes the storage requires replacement when its total throughput equals its lifetime throughput, and the battery storage approaches its required replacement. HOMER calculates the storage wear cost using the following equation:

$$C_{bw} = \frac{C_{rep,batt}}{N_{batt} \times Q_{lifetime} \times \sqrt{\eta_{rt}}},$$ (13)

where:

$C_{bw}$ = the battery wear cost ($);
$C_{rep,batt}$ = the replacement cost of the storage ($);
$N_{batt}$ = the number of batteries in the storage;
$Q_{lifetime}$ = the operating lifetime throughput of a single storage (kWh);
$\eta_{rt}$ = storage roundtrip efficiency (fractional).

### 3.4. Power Conversion System (PCS)

A power conversion system is a two-way converter (DC to AC and AC to DC). According to the requirements of charging/discharging control, PCS can convert AC power to DC power, and then store it in an energy storage battery or store energy. In addition, the DC power of the battery also can be converted into AC power to be used by the AC load. In this simulation analysis, the conversion efficiency is set to 97.3%, and the working life is set to 10 years.

### 3.5. System Dispatch Strategy

When the PV system and BESS system are set on the client-side, the simulation analysis of the HOMER Grid software trends that the load will give priority to consume the power of the PV system. The electricity for BESS charging comes from the excess electricity of the PV system and the main electricity during the period of low electricity price. When the SOC of BESS is greater than or equal to 20%, it can be discharged during the period of high electricity price. When the load demand is greater than or equal to the set contract capacity limit, BESS will discharge immediately. Under the condition without exceeding the contract capacity, BESS management to meet the electricity demand is to reduce the cost of contract capacity. The discharge timing of BESS is when the SOC is below 20%.

## 4. Component Cost and Financial Assumption

### 4.1. System Component Cost

For the simulation analysis, the cost information of required components in this case is provided by the Taiwan system integration company, as shown in Table 4. The capital cost of the PV system is $1.31 \times 10^3$/kW including PV module, PV inverter, mounting hardware, and other balance of system costs. The annual operation and maintenance (O&M) cost is 1% of PV system capital costs, with 0.25 MWp as a level for simulation. For the battery system, each three battery cabinets (639 kWh) are used as the simulation level. The price per kWh is $290/kWh, which includes DC and AC switchboards, fire protection equipment, air conditioning systems, lighting systems and energy management systems, and 20-foot containers. O&M cost per kWh per year is 1.5% of storage system capital cost, and replacement cost to the total system is 60% of storage system capital cost which is considered based on once replacement in 20 years. For PCS part, the simulation setting and analysis is relative to 0.25 MWp as each level with the price of $123/kW, and then the replacement cost to the total system is 60% capital cost which is considered based on once replacement in 20 years. The above equipment prices include transportation and installation costs. The project lifetime of the simulation analysis in this study is set to 20 years, and the residual value of the equipment is not included in the calculation.

**Table 4.** Summary of component costs.

| Description | Data Description |
| --- | --- |
| PV System | |
| Capital cost ($/kW) | 1310 |
| Operation and maintenance cost ($/kW/year) | 13.1 |
| Storage System | |
| Capital cost ($/kWh) | 290 |
| Replacement cost ($/kWh) | 174 |
| Operation and maintenance cost ($/kWh/year) | 4.35 |
| Power Conversion System | |
| Capital cost ($/kW) | 123 |
| Replacement cost (US$/kW) | 74 |
| Operation and maintenance cost (US$/kW/year) | 0 |

### 4.2. Interest Rate and Inflation Rate

According to statistics of interest rate data announced by the Central Bank of the Republic of China (Taiwan) in the past five years, the five-year average is 1.11%, and the highest and lowest are 1.36% and 1.04%, respectively. It can be seen that Taiwan has long been a low-interest-rate environment. The five-year average, as well as highest, and lowest rates are 0.73%, 2.41%, and $-0.94\%$ respectively, through the past five years statistics of the inflation rate data from Directorate General of Budget, Department of Accounting and Statistics, Executive Yuan, ROC of Taiwan's Executive Yuan. It shows that the price in Taiwan is stable for the long-term. The simulation analysis in this paper uses the average value of the statistical period as the analysis parameter, and the real discount rate is 0.38% as calculated by Equation (1).

### 4.3. Conditions without PV/BESS Installation

According to the set simulation parameters, the benchmarking comparison uses the HOMER Grid software to analyze the 20-year electricity expenditure cost under the condition of no PV/BESS installed, as the reference basis for subsequent analysis of the economic benefit evaluation after installing PV/BESS. The simulation results show the total amount of NPC in 20 years is –$54.813 \times 10^6$ which is the total electricity expenditure in 20 years and the LOCE is $0.094/kWh. The NPC in each year is shown in Figure 9.

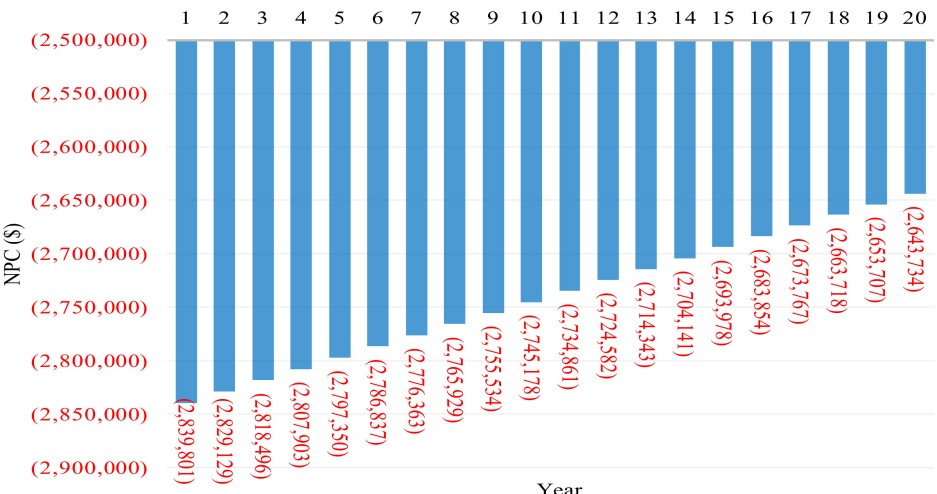

**Figure 9.** NPC in each year without PV/BESS conditions.

*4.4. Conditions with PV/BESS Installation*

Because of the installed PV system, the self-use type fully used can reduce the consumption of main power. The energy storage system uses energy arbitrage to obtain income through charge and discharge scheduling of TOU. The arbitrage method is to reduce the user contract capacity by charging electricity at off-peak electricity price and discharging electricity at peak electricity price to reduce the total electricity expense of users. To find the best project benefit of the PV system and BESS capacity, this simulation analyzes that the contract capacity is reduced from 7.5 MW to 5 MW, with each reduction level of 0.25 MW in the condition of excess electricity fraction less than 3%. After the analysis of each level is completed, the project benefit of the contract capacity of each level is compared with each other to find the reduced contract capacity with the highest project benefit and corresponding PV/BESS capacity combination. Table 5 shows the analyzed results of PV/BESS combinations corresponding to different contract capacities and the best plan is the contracted capacity of 6 MW after downgrading by the solution with a PV capacity of 8.25 MWp and the BESS capacity of 1.25 MW/3.195 MWh. The maximum project benefit of $3.591 \times 10^6$ can be obtained by resulting in the lowest LOCE of $87.85 \times 10^{-3}$/kWh, NPC of $-\$51.222 \times 10^6$, IRR of 3.08%, ROI of 1.77%, discounted payback of 15.41 years, reduced electricity charge of $18.449 \times 10^6$, total electricity savings of 33.66%, RF of 29.6%, and excess electricity fraction of 2.93%. The total setup cost of PV/BESS presents $11.888 \times 10^6$ of which PV system setup cost is $10.807 \times 10^6$ accounting for a ratio of 90.91%, and BESS's total setup cost is $1.08 \times 10^6$ accounting for a ratio of 9.09%. The total O&M cost of PV/BESS equipment in 20 years is $2.88 \times 10^6$ of which PV system is $2.078 \times 10^6$ accounting for a ratio of 72.14%, and BESS is $0.802 \times 10^6$ accounting for a ratio of 27.86%.

Figure 10 shows that when the contracted capacity reduces, more electricity charges can be saved. However, because the required capacity for PV/BESS increases, the equipment and maintenance costs will accompany by the increase to have lower project benefits. For example, the contract capacity sets at 5 MW which means the upper limit capacity of the utility power is limited to 5 MW. Therefore, the load maintain can let the system operate normally with the requirements of PV system capacity of 9 MWp and the BESS capacity of 2.25 MW/8.307 MWh, as shown in Table 5. Although the RF of contract capacity of 5 MW increases to 32.24%, the initial investment cost of the equipment has also increased by 1.22 times compared with the contract capacity of 6 MW, and the lower performance presents that the electricity charge is only reduction ratio of about 7% and the project benefit is $3.32 \times 10^6$. Although the installed capacity of the PV system is 7.25 MWp for the contract capacity at 7.25 MW, the best project benefit can be obtained with related to the lowest initial investment cost of the equipment in Table 5. The RF, electricity cost saving percentage and project benefit are all the lowest.

**Table 5.** Analyzed results of PV/BESS combinations corresponding to different contract capacities.

| Contract Capacity (kW) | 7500 | 7250 | 7000 | 6750 | 6500 | 6250 | 6000 | 5750 | 5500 | 5250 | 5000 |
|---|---|---|---|---|---|---|---|---|---|---|---|
| PV system capacity alternative | Without PV | 7250 kWp | 7500 kWp | 7750 kWp | 7750 kWp | 7750 kWp | 8250 kWp | 8250 kWp | 8250 kWp | 8500 kWp | 9000 kWp |
| BESS capacity alternative | Without BESS | 0 | 250 kW/ 639 kWh | 500 kW/ 1278 kWh | 750 kW/ 1917 kWh | 1000 kW/ 2556 kWh | 1250 kW/ 3195 kWh | 1500 kW/ 4473 kWh | 1500 kW/ 5751 kWh | 2000 kW/ 7029 kWh | 2250 kW/ 8307 kWh |
| PV system Installation Cost ($) | N/A | −10,152,500 | −9,825,000 | −10,152,500 | −10,152,500 | −10,480,000 | −10,807,500 | −10,807,500 | −10,807,500 | −11,135,000 | −11,790,000 |
| PV system maintenance cost ($) | N/A | −1,952,246 | −1,889,273 | −1,952,246 | −1,952,246 | −2,015,224 | −2,078,198 | −2,078,198 | −2,078,198 | −2,141,176 | −2,267,127 |
| BESS Installation Cost ($) | N/A | 0 | −216,110 | −432,220 | −648,330 | −864,440 | −1,080,550 | −1,481,970 | −1,883,390 | −2,284,810 | −2,686,230 |
| BESS maintenance cost ($) | N/A | 0 | −160,527 | −321,056 | −481,584 | −642,110 | −802,638 | −1,123,694 | −1,444,751 | −1,765,804 | −2,086,860 |
| Electricity expenses ($) | −54,813,205 | −39,914,431 | −39,752,863 | −38,812,925 | −38,283,773 | −37,362,064 | −36,453,117 | −35,765,228 | −35,096,604 | −34,048,495 | −32,662,915 |
| NPC ($) | −54,813,205 | −52,019,181 | −51,843,772 | −51,670,949 | −51,518,438 | −51,363,840 | −51,222,008 | −51,256,593 | −51,310,443 | −51,375,287 | −51,493,136 |
| Discounted electricity cost saving ($) | N/A | 14,898,774 | 15,078,139 | 16,035,874 | 16,582,823 | 17,522,329 | 18,449,073 | 19,154,760 | 19,841,181 | 20,907,087 | 22,310,464 |
| Project benefit ($) | N/A | 2,794,024 | 2,969,433 | 3,142,256 | 3,294,767 | 3,449,365 | 3,591,197 | 3,556,612 | 3,502,762 | 3,437,918 | 3,320,069 |
| Return on investment (%) | N/A | 1.63 | 1.74 | 1.74 | 1.79 | 1.78 | 1.77 | 1.71 | 1.64 | 1.53 | 1.39 |
| Internal rate of return (%) | N/A | 2.85 | 3.03 | 3.04 | 3.11 | 3.10 | 3.08 | 2.97 | 2.86 | 2.69 | 2.46 |
| Discounted payback (year) | N/A | 15.55 | 15.35 | 15.37 | 15.31 | 15.36 | 15.41 | 15.61 | 15.82 | 16.10 | 16.46 |
| LOCE ($/kWh) | 0.09401 | 0.08921 | 0.08891 | 0.08862 | 0.08835 | 0.08809 | 0.08785 | 0.08791 | 0.08800 | 0.08811 | 0.08831 |
| Electricity cost saving (%) | 0 | 27.18 | 27.51 | 29.26 | 30.25 | 31.97 | 33.66 | 34.95 | 36.20 | 38.14 | 40.70 |
| Excess electricity fraction (%) | N/A | 2.86 | 2.90 | 2.96 | 2.77 | 2.85 | 2.93 | 2.64 | 2.39 | 2.37 | 2.57 |
| Renewable fraction (%) | N/A | 26.10 | 27.00 | 27.90 | 28.00 | 28.80 | 29.60 | 29.80 | 29.90 | 30.80 | 32.40 |

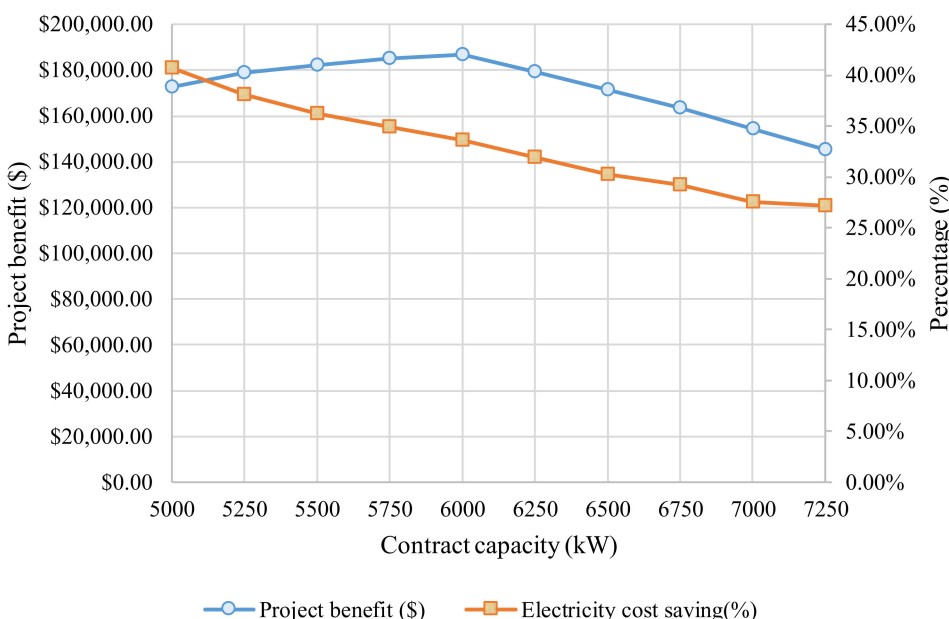

**Figure 10.** The project benefit and the electricity saving percentage of PV/BESS combination corresponding to different contract capacity.

As an example to illustrate the electricity consumption in summer, it can be presented by taking the contracted capacity of 6 MW, PV capacity of 8.25 MWp, and BESS capacity of 1.25 MW/3.195 MWh. To TOU management, BESS charged during the off-peak period with low electricity price on 18 June as shown in Figure 11. The load power consumption appears at 6.179 MW, which is higher than the setting value of contract capacity, in the period with a high electricity price at 10:00 o'clock on the next day. Because of the energy management of power supply by PV system outputs of 3.856 MW and the BESS outputs of 1.25 MW, the immediate usage of main power drops to 1.073 MW. On the same day, the consumption of main power can be as low to 0 MW. In addition to meeting power demand, the solution can reduce main power consumption and electricity charge related to avoiding high fines due to load usage exceeding contract capacity.

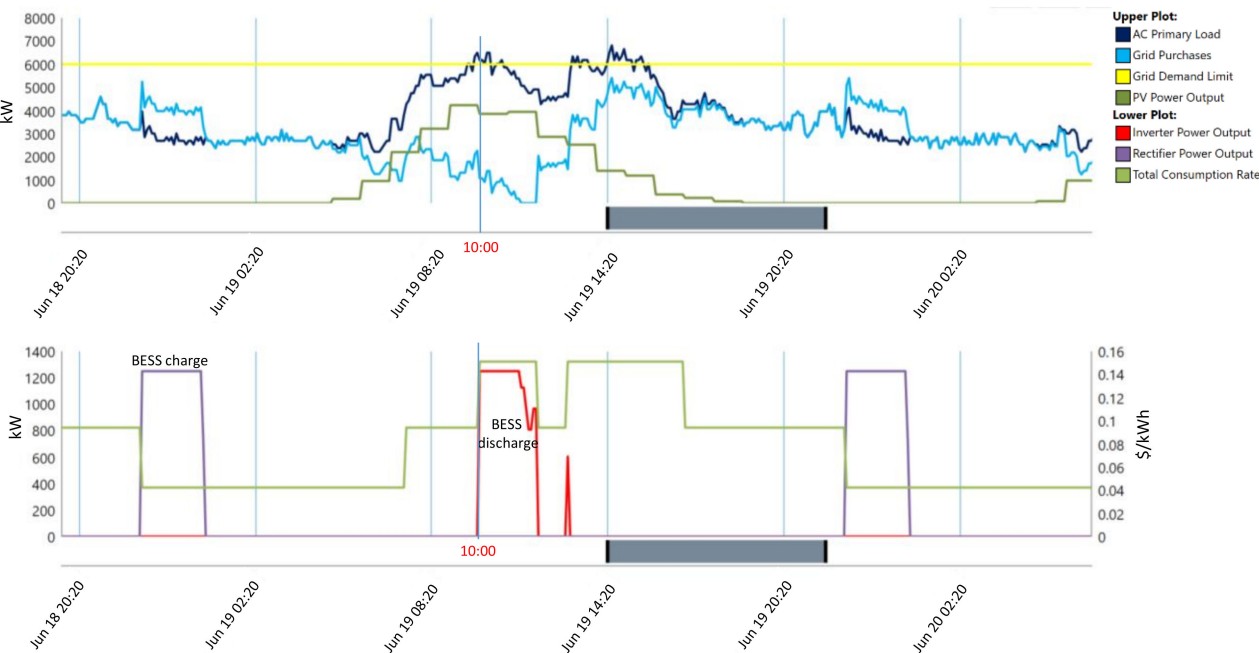

**Figure 11.** The power profile of PV/BESS in the BTM application.

## 5. Sensitivity Analysis

Base on the analysis results in Section 5.2, the parameters of global horizontal irradiation, electricity price, BESS capital cost, and real interest rate were simulated to different levels of user contract capacity to determine the best project benefit and PV/BESS scheme.

### 5.1. Global Horizontal Irradiation

To sensitivity analysis, the simulation is set by GHI ranging from 3.4 to 5.0 kW/m²/day with the interval of 0.2 kW/m²/day. When the GHI increases, RF increases and grid purchases decrease in Figure 12, and the NPC decreases accordingly in Figure 13. When the expenditure of cash flow decreases, the project benefit will increase, and the LCOE and discounted payback will both decrease as shown in Figures 13 and 14. According to GHI of 5.0 kW/m²/day, it results that RF rises to 34.6%, NPC drops to $-\$48.679 \times 10^6$, project benefit increases to $\$6.134 \times 10^6$, LCOE drops to $\$83.49 \times 10^{-3}$/kWh, and discounted payback is reduced to 13.25 years. The longer sunshine time leads to higher investment benefits.

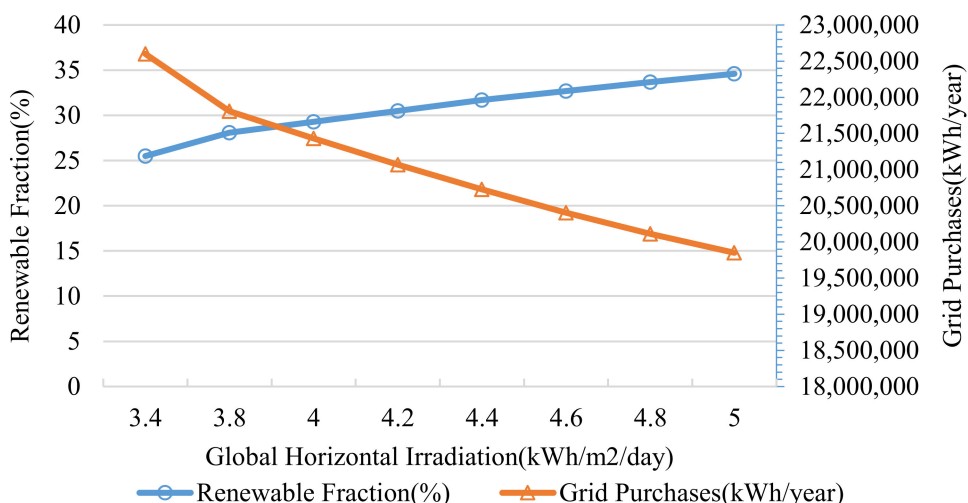

**Figure 12.** Effect of varying on the RF and grid purchases.

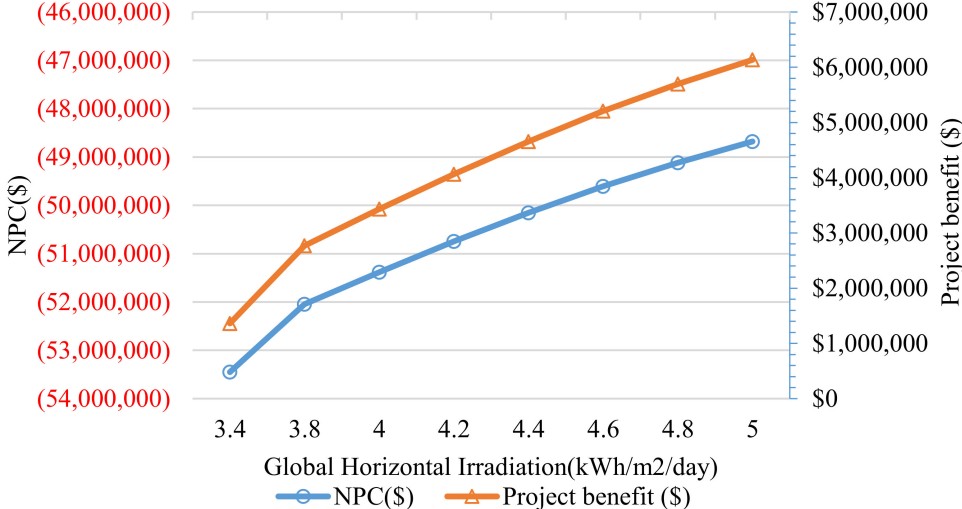

**Figure 13.** Effect of varying GHI on the NPC and project benefit.

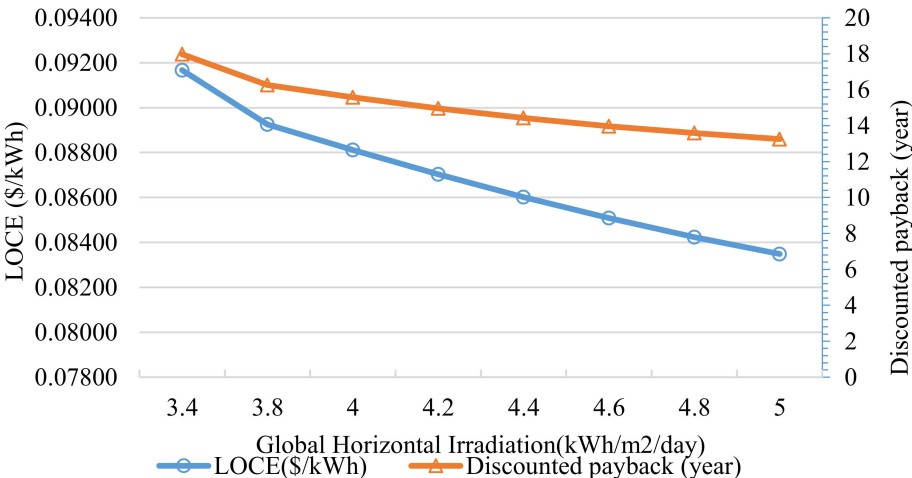

**Figure 14.** Effect of varying GHI on the LCOE and discounted payback.

### 5.2. Conditions without PV/BESS Installation

According to the set simulation parameters, the benchmarking comparison uses the HOMER Grid software to analyze the 20-year electricity expenditure cost under the condition of no PV/BESS installed, as the reference basis for subsequent analysis of the economic benefit evaluation after installing PV/BESS. The simulation results show the total amount of NPC in 20 years is $-\$54.813 \times 10^6$, which is the total electricity expenditure in 20 years and the LOCE is $\$94.01 \times 10^{-3}$/kWh. The NPC in each year is shown in Figure 9.

### 5.3. Electricity Price

According to the electricity price regulations in Taiwan Electric Power Company, the maximum range of each electricity price adjustment is 3%. Therefore, the sensitivity analysis of this study is based on the comparison and analysis of the electricity price for four cases, without adjustment for 20 years, 3% adjustment every three years, 3% adjustment every two years, and 3% adjustment every year. According to the analysis results of Table 6, the electricity price adjusted more frequently leads to higher accumulative electricity bills in 20 years including both increased NPC and LCOE. However, the Project benefit, ROI, and IRR will increase, and the discounted payback will be shortened under the condition of PV/BESS. As an example by taking the scenario of adjusting 3% every three years, LOCE will be as high as $0.12044/kWh without PV/BESS installation, but LOCE will be reduced to $0.10539/kWh and IRR will be increased to 5.87% with PV/BESS installation. In the context of high electricity prices, the results show the benefits of installing PV/BESS can be more prominent, even in effect for Taiwan's electricity prices at low levels in the world.

**Table 6.** Analyzed results of PV/BESS combinations corresponding to different contract capacities.

| Types of Electricity Price Rates for Different Years | 10 Years of Electricity Price Rate Unchanged | Increased by 3% Every Three Years | Increased by 3% Every Two Years | Increase by 3% Every Year |
|---|---|---|---|---|
| NPC-Without PV/BESS ($) | −$54,813,205 | −$59,431,490 | −$62,110,835 | −$70,229,116 |
| NPC-With PV/BESS ($) | −$51,222,008 | −$54,285,866 | −$56,063,396 | −$61,449,217 |
| Project benefit ($) | $3591,197 | $5,145,624 | $6,047,439 | $8,779,899 |
| ROI (%) | 1.77% | 2.46% | 2.86% | 4.07% |
| IRR (%) | 3.08% | 4.00% | 4.50% | 5.87% |
| Discounted payback (year) | 15.41 | 14.44 | 13.95 | 13.95 |
| LOCE-Without PV/BESS ($/kWh) | 0.09401 | 0.10193 | 0.10652 | 0.12044 |
| LOCE-With PV/BESS ($/kWh) | 0.08785 | 0.09310 | 0.09615 | 0.10539 |

### 5.4. PV/BESS Capital Cost

The sensitivity analysis of PV/BESS capital cost is divided into four parts: total PV/BESS capital cost, PV system capital cost, BESS capital cost, and PCS capital cost. The evaluation analyzes the impact on economic benefits under different cost conditions with a change ratio of 1–0.5 times and each interval of 10%. Figure 15 shows the economic benefits obtained are significantly higher than reducing the BESS or PCS capital cost by reducing the expenditure of the PV system capital cost. Because the capital cost of the PV system is higher than that of BESS and PCS in terms of the degree of impact on economic benefits. According to the analyzed result of Table 7, the project benefit can be increased by 30.09% if the expenditure of PV system capital cost is reduced by 10%. However, the project benefits will restrictively increase by 4.07 and 0.68% related to capital cost expenditure of BESS or PCS reduced by 10%, respectively. If the expenditure of total PV/BESS capital cost is reduced by 10%, the results at the ratio of 0.9 reveal that the project benefit increased to 34.84%, the IRR increased to 4.30%, and the discounted payback shortened to 13.83 years.

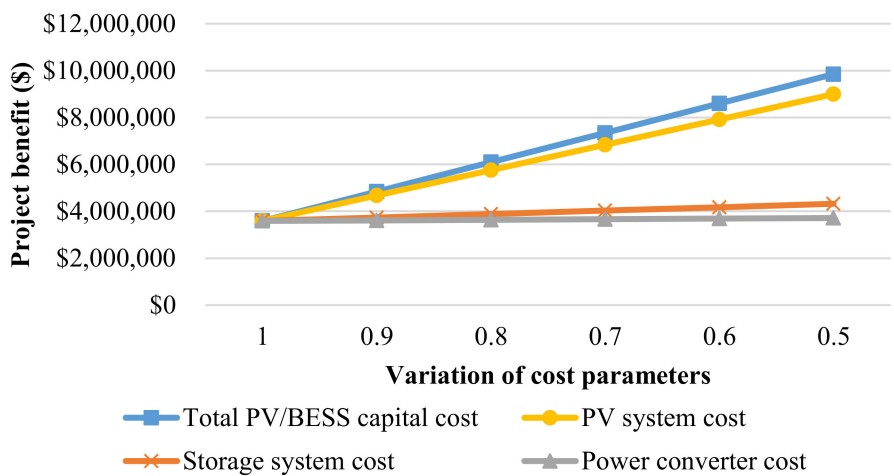

**Figure 15.** Effect of varying key capital costs on the project benefit.

**Table 7.** Analyzed results of PV/BESS combinations corresponding to different contract capacities.

| PV/BESS Cost Reduction Ratio | 1 | 0.9 | 0.8 | 0.7 | 0.6 | 0.5 |
|---|---|---|---|---|---|---|
| NPC-Without PV/BESS ($) | −$54,813,205 | −$54,813,205 | −$54,813,205 | −$54,813,205 | −$54,813,205 | −$54,813,205 |
| NPC-With PV/BESS ($) | −$51,222,008 | −$49,970,765 | −$48,719,523 | −$47,468,281 | −$46,217,039 | −$44,965,797 |
| Project benefit ($) | $3,591,197 | $4,842,440 | $6,093,682 | $7,344,924 | $8,596,166 | $9,847,408 |
| ROI (%) | 1.77% | 2.55% | 3.53% | 4.79% | 6.47% | 8.82% |
| IRR(%) | 3.08% | 4.30% | 5.75% | 7.49% | 9.68% | 12.56% |
| Discounted payback (year) | 15.41 | 13.83 | 12.25 | 10.69 | 8.67 | 7.21 |
| LOCE-Without PV/BESS ($/kWh) | 0.09401 | 0.09401 | 0.09401 | 0.09401 | 0.09401 | 0.09401 |
| LOCE-With PV/BESS ($/kWh) | 0.08785 | 0.08570 | 0.08355 | 0.08141 | 0.07926 | 0.07712 |
| PV cost reduction ratio | 1 | 0.9 | 0.8 | 0.7 | 0.6 | 0.5 |
| NPC-Without PV/BESS ($) | −$54,813,205 | −$54,813,205 | −$54,813,205 | −$54,813,205 | −$54,813,205 | −$54,813,205 |
| NPC-With PV/BESS ($) | −$51,222,008 | −$50,141,258 | −$49,060,508 | −$47,979,758 | −$46,899,008 | −$45,818,258 |
| Project benefit ($) | $3,591,197 | $4,671,947 | $5,752,697 | $6,833,447 | $7,914,197 | $8,994,947 |

**Table 7.** *Cont.*

| PV/BESS Cost Reduction Ratio | 1 | 0.9 | 0.8 | 0.7 | 0.6 | 0.5 |
|---|---|---|---|---|---|---|
| ROI (%) | 1.77% | 2.45% | 3.28% | 4.31% | 5.64% | 7.42% |
| IRR (%) | 3.08% | 4.14% | 5.38% | 6.84% | 8.63% | 10.90% |
| Discounted payback (year) | 15.41 | 14.04 | 12.68 | 11.33 | 9.92 | 7.87 |
| LOCE-Without PV/BESS ($/kWh) | 0.09401 | 0.09401 | 0.09401 | 0.09401 | 0.09401 | 0.09401 |
| LOCE-With PV/BESS ($/kWh) | 0.08785 | 0.08599 | 0.08414 | 0.08229 | 0.08043 | 0.07858 |
| Storage system cost reduction ratio | 1 | 0.9 | 0.8 | 0.7 | 0.6 | 0.5 |
| NPC-Without PV/BESS ($) | −$54,813,205 | −$54,813,205 | −$54,813,205 | −$54,813,205 | −$54,813,205 | −$54,813,205 |
| NPC-With PV/BESS($) | −$51,222,008 | −$51,075,814 | −$50,929,620 | −$50,783,427 | −$50,637,233 | −$50,491,039 |
| Project benefit ($) | $3,591,197 | $3,737,391 | $3,883,585 | $4,029,778 | $4,175,972 | $4,322,166 |
| ROI (%) | 1.77% | 1.85% | 1.93% | 2.01% | 2.09% | 2.17% |
| IRR (%) | 3.08% | 3.20% | 3.33% | 3.45% | 3.58% | 3.71% |
| Discounted payback (year) | 15.41 | 15.22 | 15.04 | 14.85 | 14.67 | 14.48 |
| LOCE-Without PV/BESS ($/kWh) | 0.09401 | 0.09401 | 0.09401 | 0.09401 | 0.09401 | 0.09401 |
| LOCE-With PV/BESS ($/kWh) | 0.08785 | 0.08760 | 0.08735 | 0.08709 | 0.08684 | 0.08659 |
| PCS cost reduction ratio | 1 | 0.9 | 0.8 | 0.7 | 0.6 | 0.5 |
| NPC-Without PV/BESS ($) | −$54,813,205 | −$54,813,205 | −$54,813,205 | −$54,813,205 | −$54,813,205 | −$54,813,205 |
| NPC-With PV/BESS ($) | −$51,222,008 | −$51,197,709 | −$51,173,410 | −$51,149,112 | −$51,124,813 | −$51,100,515 |
| Project benefit ($) | $3,591,197 | $3,615,496 | $3,639,795 | $3,664,093 | $3,688,392 | $3,712,690 |
| ROI (%) | 1.77% | 1.78% | 1.80% | 1.81% | 1.82% | 1.84% |
| IRR (%) | 3.08% | 3.10% | 3.12% | 3.14% | 3.16% | 3.18% |
| Discounted payback (year) | 15.41 | 15.38 | 15.35 | 15.32 | 15.28 | 15.25 |
| LOCE-Without PV/BESS ($/kWh) | 0.09401 | 0.09401 | 0.09401 | 0.09401 | 0.09401 | 0.09401 |
| LOCE-With PV/BESS ($/kWh) | 0.08785 | 0.08780 | 0.08776 | 0.08772 | 0.08768 | 0.08764 |

### 5.5. Real Interest Rate

This analysis sets the real interest rate changes from −5% to 5% with intervals of 1%. From Equation (1) to Equation (5) and the analysis results shown in Figures 16 and 17, it can be seen that a higher real interest rate leads to the lower present value of cash flows each year in the future. NPC will be seen as decreased, but the project benefit of PV/BESS will be decreased when the real interest rate increases. Therefore, LOCE and discounted payback will increase with the increase of the real interest rate. The analysis results of ROI and IRR can also be obtained from Equations (8) and (9), which are calculated by using the annual cash flow value without considering real interest rate, so it is not relative to real interest rate changes. In Figure 16 no more data point of discounted payback at an interest rate of 4% shows the 20-year payment with the PV/BESS scenario is much more than that without PV/BESS, so the project benefit is negatively related to the recovery period already more than 20 years.

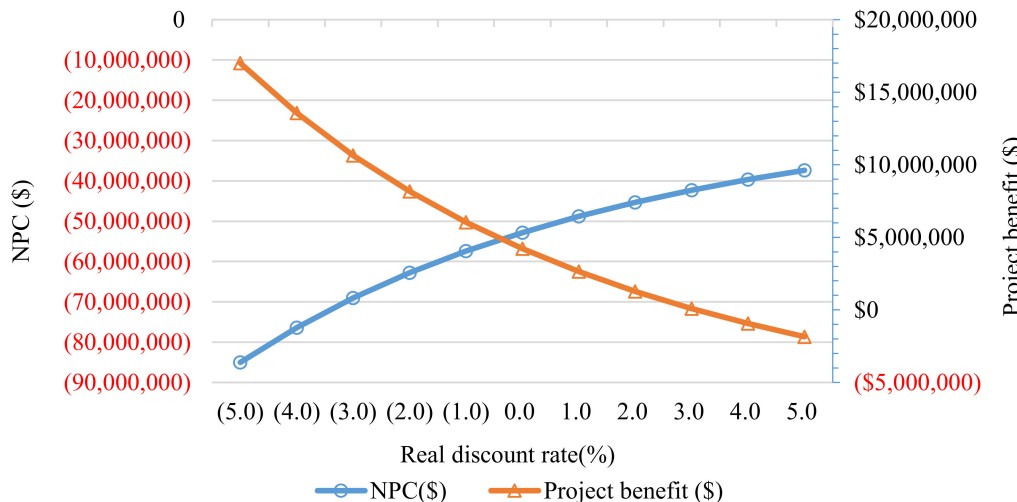

**Figure 16.** Effect of varying real interest rates on the NPC and project benefit.

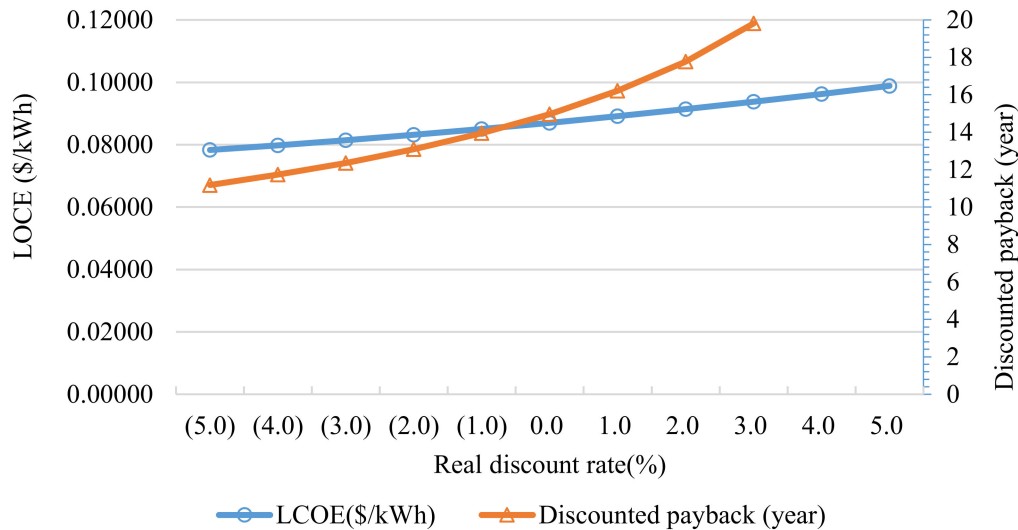

**Figure 17.** Effect of varying real interest rates on the LCOE and discounted payback.

## 6. Conclusions

According to the case study and simulation analysis in this research, the techno-economic analysis of PV/BESS in BTM application can be performed through the HOMER Grid software to find the PV/BESS capacity configuration with the highest economic benefit, and the corresponding contract capacity scheme. This method proposed the applications to PV/BESS economic benefit analysis of different load profiles, TOU modes, etc. The comprehensive simulation analysis results of this research are as follows:

According to the load conditions, the analysis includes electricity price data, PV/BESS equipment and O&M costs, limit conditions, and trends no PV/BESS installation as a benchmark for comparison. Through the HOMER Grid software, a combination scheme can be simulated and analyzed by resulting in the most economical solution with the capacity of a PV system of 8.25 MWp, BESS of 1.25 MW/3.195 MWh, and the corresponding contract capacity of 6 MW. The benefits for the economical solution are presented as 20-year project benefit of $3.591 \times 10^6$, LOCE of $87.85 \times 10^{-3}$/kWh, IRR of 3.08%, ROI of 1.77%, discounted payback of 15.41 years, total electricity savings of 33.66%, RF of 29.6%, and excess electricity fraction of 2.93%. When the PV/BESS capacity is larger, the corresponding adjustable contract capacity is larger than the saved electricity charge. However, the investment cost of equipment and maintenance cost also increase, which reduces the project benefit. Based on sensitivity analysis to the highest project benefit

scheme, the context of higher sunshine conditions and electricity prices leads to higher investment benefit, energy arbitrage benefit, ROI, IRR, and shorter discounted payback to reveal the benefits of installing PV/BESS.

In the sensitivity analysis on cost, the capital cost ratio of the PV system is higher than that of BESS and PCS. In terms of the impact on economic benefits, the economic benefits obtained by reducing the capital cost of the PV system are significantly higher than reducing the capital costs of BESS and PCS. For the sensitivity analysis on the real interest rate, the increased real interest rate corresponds to decreased NPC, decreased project benefit, increased LOCE and longer discounted payback. Therefore, the context of low real interest rate is significant to the economic benefits of PV/BESS installation.

**Author Contributions:** Conceptualization, C.-Y.P. and C.-C.K.; Formal analysis, C.-Y.P. and C.-T.T.; Funding acquisition, C.-C.K.; Investigation, C.-Y.P.; Methodology, C.-Y.P. and C.-T.T.; Project administration, C.-C.K.; Software, C.-T.T.; Validation, C.-Y.P. and C.-T.T.; Visualization, C.-T.T.; Writing—original draft, C.-Y.P. and C.-C.K.; Writing—review & editing, C.-C.K. All authors have read and agreed to the published version of the manuscript.

**Funding:** The financial support for Energy Project provided by Taiwan Bureau of Energy is gratefully acknowledged according to Energy Storage System and Distributed Smart Microgrid Development Project for DC 750 V High Voltage.

**Institutional Review Board Statement:** Not applicable.

**Informed Consent Statement:** Not applicable.

**Data Availability Statement:** Not applicable.

**Conflicts of Interest:** The authors declare no conflict of interest.

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
