# Peer review of "Optimal Configuration with Capacity Analysis of PV-Plus-BESS for Behind-the-Meter Application"

_applsci, doi:10.3390/app11177851_

Round 1

Reviewer 1 Report

We cannot consider that this paper is a research paper ; the authors used a commercial software « HOMER » and realized several simulations. They explained all the equation incorporated into this software and we cannot consider that there is a novelty in this paper. Moreover, several points are not clear at all. Try to add some novelties and to apply an optimization method.

Line 165 : please add the unity of latitude and longitude (°)

Lines 171-174 : can be deleted ; it is easy to see it in Figure

Figure 4 and line 194 : No it is impossible to see the daily load profile in Fig.4. Replace this figure and plot the daily profile where it will be possible to see the peak period and the power over the day.

Lines 389-390 : we don’t understand the C-rate ? what does it means ? the C-rate can be limited but it varies with the load power and the PV power ; it is not clear ! it is not clear too the sentence «  lifetime throughout is 967,765kWh » we don’t understand too.

Line 419 : the replacement cost is 60% capital cost : be clearer in your hypothesis ; the replacement cost is added at the end of the battery lifetime, yes or no ? do you applied an actualization rate ? capital cost : which capital cost is ? of the total system ? of the battery ? how many times do you add the replacement cost ? which is the battery lifetime ?

Line 423 : the system lifetime is 20 years : could you justify ? today we can consider that a pV lifetime is 30 years ; how did you chose the lifetime ? the contract with the electricity supplier is for how many years ?

Line 506 : The method is not clearly described

Try to be homogeneous; sometimes, you used "$" and "US$",

The number are difficult to read  as an ex : $6,047,439.00. try to change the units;  units not well chosen  ex : 21,500,000 kWh

Author Response

Reply to reviewer’s comments:

According to reviewer’s comments the revisions had been completed using the “track changes with any change parts”. The cover letter to each reviewer provides the explanations, addressing certain comments with point by point responses to the referees’ comments.

Reviewer 1

Point 1: We cannot consider that this paper is a research paper ; the authors used a commercial software « HOMER » and realized several simulations. They explained all the equation incorporated into this software and we cannot consider that there is a novelty in this paper. Moreover, several points are not clear at all. Try to add some novelties and to apply an optimization method.

Response 1:

For this comment we already add (1) the focus, (2) Taiwan’s policy driving (new 4 references) and (3) the flowchart in figure 1 as shown in the following:

(1) the focus in the introduction: the excess electricity fraction is limited to 3% to a novel precise analysis and effective use of energy for the economic benefit evaluation of PV/BESS.

Based on Taiwan's energy policy, this article mainly discusses the economic benefits of integrating the PV system with BESS and applying BTM. Use HOMER Grid software to conduct electrical and economic simulation analysis with actual cases. According to the previous economic analysis of PV or PV/BESS works of literature, the traditional methods directly apply the load demand or electricity price evaluations to set PV capacity, but the constraints of the excess electricity fraction are less considered to the correspondence be-tween renewable fraction (RF) and exces power. In usually the operation of excess power uses power curtailment by the solar inverter and reveals a waste of excess PV power. Therefore, the excess electricity fraction is limited to 3% to a novel precise analysis and effective use of energy for the economic benefit evaluation of PV/BESS. The energy arbitrage revenue can be obtained by calculating the optimal economical capacity allocation of a PV system and BESS in the conditions of meet load demand and PV’s RF constraints at the same time.

Please refer to the introduction section, line 139 to 151.

(2) the Taiwan’s policy driving in the introduction:

Since Taiwan passed the amendment to the "Renewable Energy Development Act" in 2019, the policy tools strongly promote the energy development of the industry [29-31]. According to article 12, the electricity consumers above a certain contracted capacity must be stipulated by one of the requirements in the following: (1) install a certain proportion of renewable energy equipment for power generation, (2) install a battery energy storage sys-tem, (3) purchase a certain proportion of green power of Taiwan Renewable Energy Certif-icate (T-REC) or (4) pay the fine. The primary specification object is set on the electricity consumer with a contract capacity of 5,000kW or more by the implementation rules with the obligation fulfilled within 5 years. After the implementation, a rolling review every two years will gradually expand the primary specification object. The requirement presents 10% of the contract capacity to the installed capacity of renewable energy equipment for power generation while the BESS's capacity meets 2 hours [32].

Please refer to the introduction section, line 127 to 138.

Please refer to the references [29-32]

(3) the analysis flowchart in figure 1:

According to Taiwan’s policy of contracted capacity, the objective of analysis shows the reduced contracted capacity related to the most economical PV/BESS capacity alloca-tion and optimal contract capacity scheme under the condition that the electricity demand is met and the PV power generation is fully used. Figure 1 shows the flowchart of the eco-nomic benefit analysis. Without a waste of excess PV power, the excess electricity fraction is limited within 3% to the analysis and effective use of energy for the economic benefit evaluation of PV/BESS.

Please refer to line 163 to 171.

Point 2: Line 165 : please add the unity of latitude and longitude (°)

Response 2:

For this comment, we have revised the manuscript.

Please refer to the abstract section, line 211.

Point 3: Lines 171-174 : can be deleted ; it is easy to see it in Figure

Response 3:

For this comment, we have deleted the paragraph in the manuscript.

Please refer to line 217 to 220.

Point 4: Figure 4 and line 194 : No it is impossible to see the daily load profile in Fig.4. Replace this figure and plot the daily profile where it will be possible to see the peak period and the power over the day.

Response 4:

For this comment, we have updated the daily load profile in the manuscript.

Figure 5. The load profile for the average daily power consumption of annual energy.

Please refer to line 254 to 257.

Point 5: Lines 389-390 : we don’t understand the C-rate ? what does it means ? the C-rate can be limited but it varies with the load power and the PV power ; it is not clear ! it is not clear too the sentence «  lifetime throughout is 967,765kWh » we don’t understand too.

Response 5:

For this comment, we have improved and revised the manuscript.

C rate presents as a normalized battery capacity by being charge and discharge currents and as discharged relative to its maximum capacity. A 0.5C rate of the battery is applied to the storage system that means he discharge current of 50 Amps and discharge the entire battery in 2 hours. The lifetime presents the number of discharge-charge cycles to meet specific performance criteria of the battery. The operating lifetime throughout of 967,765kWh of the battery is affected by the parameters (the rate and depth of cycles) and by the environmental conditions (temperature and humidity). The warranty of a commercial battery is the ranges from 5-10 years according to industrial experiences. The related standard such as IEC62620 can provide the parameters to lifetime according to the customer’s declarations.

Please refer to line 447 to 456.

Point 6: Line 419 : the replacement cost is 60% capital cost : be clearer in your hypothesis ; the replacement cost is added at the end of the battery lifetime, yes or no ? do you applied an actualization rate ? capital cost : which capital cost is ? of the total system ? of the battery ? how many times do you add the replacement cost ? which is the battery lifetime ?

Response 6:

For this comment, we have improved and revised the manuscript.

O&M cost per kWh per year is 1.5% of storage system capital cost, and replacement cost to total system is 60% of storage system capital cost which is considered based on once replacement in 20 years. For PCS part, the simulation setting and analysis is relative to 250kWp as each level with the price of $123/kW, and then the replacement cost to total system is 60% capital cost which is considered based on once replacement in 20 years. The above equipment prices include transportation and installation costs. The project life time of the simulation analysis in this study is set to 20 years, and the residual value of the equipment is not included in the calculation.

Please refer to line 485 to 492.

Point 7: Line 423 : the system lifetime is 20 years : could you justify ? today we can consider that a PV lifetime is 30 years ; how did you chose the lifetime ? the contract with the electricity supplier is for how many years ?

Response 7:

In this paper, we propose an analyzed method to determine the most economical PV/BESS capacity alloca-tion and optimal contract capacity scheme. It is possible to justify by changing lifetime, but this paper focuses on the system evaluation that is estimated based on 20 years according to Taiwan’s environment including policy and industrial.

Point 8: Line 506 : The method is not clearly described

Response 8:

For this comment, we have improved and revised the manuscript.

According to Taiwan’s policy of contracted capacity, the objective of analysis shows the reduced contracted capacity related to the most economical PV/BESS capacity alloca-tion and optimal contract capacity scheme under the condition that the electricity demand is met and the PV power generation is fully used. Figure 1 shows the flowchart of the eco-nomic benefit analysis. Without a waste of excess PV power, the excess electricity fraction is limited within 3% to the analysis and effective use of energy for the economic benefit evaluation of PV/BESS.

Figure 1. The flowchart of the economic benefit analysis.

Please refer to line 163 to 169.

Point 9: Try to be homogeneous; sometimes, you used "$" and "US$",

Response 9:

For this comment, we have revised the manuscript. Thanks for the reviewer’s commend.

Point 10: The number are difficult to read  as an ex : $6,047,439.00. try to change the units;  units not well chosen  ex : 21,500,000 kWh

Response 10:

For this comment, we have revised for a decimal point and contract capacity unit in MW in the manuscript, but we want to keep kWh to make the electricity and power consistent throughout the paper. Please kindly understand.

Reviewer 2 Report

This manuscript by Peng et al describes the PV-plus-BESS applied to the behind-the-meter market. The experimental results are solid and systematic. However, there are many grammar errors in text and the sentences are difficult for the readers to understand. I conclude the work can be considered to be published in Applied Sciences after extensive editing the English language and style.

Author Response

Reply to reviewer’s comments:

According to reviewer’s comments the revisions had been completed using the “track changes with any change parts”. The cover letter to each reviewer provides the explanations, addressing certain comments with point by point responses to the referees’ comments.

Reviewer 2

Point 1: This manuscript by Peng et al describes the PV-plus-BESS applied to the behind-the-meter market. The experimental results are solid and systematic. However, there are many grammar errors in text and the sentences are difficult for the readers to understand. I conclude the work can be considered to be published in Applied Sciences after extensive editing the English language and style.

Response 1:

Many thanks for the reviewer’s commend. After checking, the English writing and grammar errors have been checked and fixed by a commercial service in grammarly.com.

Reviewer 3 Report

This research uses the contract capacity of the actual industrial users of 7,500kW as a research case and simulates PV/BESS techno-economic schemes through HOMER Grid software. I have the following concerns.

  1. What is the contribution of this study? I do not find any new methods/results. Please clarify briefly.
  2. The literature section should be conducted with a critical review of existing methods. This is missing. More literature should be included.
  3. What is the knowledge gap that this has been filled? Please explain.
  4. The introduction should be organised. Please check the format of a good research paper.
  5. Please clearly shows the objective/aim of this study.
  6. Please elaborate on the significance and benefit of this study.
  7. Writing could be improved.

Author Response

Reply to reviewer’s comments:

According to reviewer’s comments the revisions had been completed using the “track changes with any change parts”. The cover letter to each reviewer provides the explanations, addressing certain comments with point by point responses to the referees’ comments.

Reviewer 3

This research uses the contract capacity of the actual industrial users of 7,500kW as a research case and simulates PV/BESS techno-economic schemes through HOMER Grid software. I have the following concerns.

Point 1: What is the contribution of this study? I do not find any new methods/results. Please clarify briefly.

Response 1:

For this comment we already add (1) the focus, (2) Taiwan’s policy driving (new 4 references) and (3) the flowchart in figure 1 as shown in the following:

(1) the focus in the introduction: the excess electricity fraction is limited to 3% to a novel precise analysis and effective use of energy for the economic benefit evaluation of PV/BESS.

Based on Taiwan's energy policy, this article mainly discusses the economic benefits of integrating the PV system with BESS and applying BTM. Use HOMER Grid software to conduct electrical and economic simulation analysis with actual cases. According to the previous economic analysis of PV or PV/BESS works of literature, the traditional methods directly apply the load demand or electricity price evaluations to set PV capacity, but the constraints of the excess electricity fraction are less considered to the correspondence be-tween renewable fraction (RF) and exces power. In usually the operation of excess power uses power curtailment by the solar inverter and reveals a waste of excess PV power. Therefore, the excess electricity fraction is limited to 3% to a novel precise analysis and effective use of energy for the economic benefit evaluation of PV/BESS. The energy arbitrage revenue can be obtained by calculating the optimal economical capacity allocation of a PV system and BESS in the conditions of meet load demand and PV’s RF constraints at the same time.

Please refer to the introduction section, line 139 to 151

(2) the Taiwan’s policy driving in the introduction:

Since Taiwan passed the amendment to the "Renewable Energy Development Act" in 2019, the policy tools strongly promote the energy development of the industry [29-31]. According to article 12, the electricity consumers above a certain contracted capacity must be stipulated by one of the requirements in the following: (1) install a certain proportion of renewable energy equipment for power generation, (2) install a battery energy storage sys-tem, (3) purchase a certain proportion of green power of Taiwan Renewable Energy Certif-icate (T-REC) or (4) pay the fine. The primary specification object is set on the electricity consumer with a contract capacity of 5,000kW or more by the implementation rules with the obligation fulfilled within 5 years. After the implementation, a rolling review every two years will gradually expand the primary specification object. The requirement presents 10% of the contract capacity to the installed capacity of renewable energy equipment for power generation while the BESS's capacity meets 2 hours [32].

Please refer to the introduction section, line 127 to 138

Please refer to the references [29-32]

Point 2: The literature section should be conducted with a critical review of existing methods. This is missing. More literature should be included.

Response 2:

For this comment we have already revised the introduction and added new 4 references Taiwan’s policy driving as shown in the following:

The Taiwan’s policy driving in the introduction:

Since Taiwan passed the amendment to the "Renewable Energy Development Act" in 2019, the policy tools strongly promote the energy development of the industry [29-31]. According to article 12, the electricity consumers above a certain contracted capacity must be stipulated by one of the requirements in the following: (1) install a certain proportion of renewable energy equipment for power generation, (2) install a battery energy storage sys-tem, (3) purchase a certain proportion of green power of Taiwan Renewable Energy Certif-icate (T-REC) or (4) pay the fine. The primary specification object is set on the electricity consumer with a contract capacity of 5MW or more by the implementation rules with the obligation fulfilled within 5 years. After the implementation, a rolling review every two years will gradually expand the primary specification object. The requirement presents 10% of the contract capacity to the installed capacity of renewable energy equipment for power generation while the BESS's capacity meets 2 hours [32].

Please refer to the introduction section, line 127 to 138

Please refer to the references [29-32]

Point 3: What is the knowledge gap that this has been filled? Please explain.

Response 3:

For this comment, we have improved and revised the manuscript.

According to Taiwan’s policy of contracted capacity, the objective of analysis shows the reduced contracted capacity related to the most economical PV/BESS capacity alloca-tion and optimal contract capacity scheme under the condition that the electricity demand is met and the PV power generation is fully used. Figure 1 shows the flowchart of the eco-nomic benefit analysis. Without a waste of excess PV power, the excess electricity fraction is limited within 3% to the analysis and effective use of energy for the economic benefit evaluation of PV/BESS.

Figure 1. The flowchart of the economic benefit analysis.

Please refer to line 163 to 171.

Point 4: The introduction should be organised. Please check the format of a good research paper.

Response 4:

For this comment we have rewrited and reorganized the introduction part.

Point 5: Please clearly shows the objective/aim of this study.

Response 5:

For this comment we have shown in the abstrate part.

As the cost of photovoltaic (PV) systems and battery energy storage systems (BESS) decreases, PV-plus-BESS applied to behind-the-meter (BTM) market has grown rapidly in recent years. With user time of use rates (TOU) for charging and discharging schedule, it can effectively reduce the electricity expense of users. This research uses the contract capacity of an actual industrial user of 7.5MW as a research case, and simulates a PV/BESS techno-economic scheme through HOMER Grid software. Under the condition that the electricity demand is met and the PV power generation is fully used, the target is to find the most economical PV/BESS capacity allocation and optimal contract capacity scheme.

Point 6: Please elaborate on the significance and benefit of this study.

Response 6:

For this comment, we already add the the significance and benefit of this study as shown in the following:

The significance and benefit of this study in the introduction: the excess electricity fraction is limited to 3% to a novel precise analysis and effective use of energy for the economic benefit evaluation of PV/BESS.

Based on Taiwan's energy policy, this article mainly discusses the economic benefits of integrating the PV system with BESS and applying BTM. Use HOMER Grid software to conduct electrical and economic simulation analysis with actual cases. According to the previous economic analysis of PV or PV/BESS works of literature, the traditional methods directly apply the load demand or electricity price evaluations to set PV capacity, but the constraints of the excess electricity fraction are less considered to the correspondence be-tween renewable fraction (RF) and exces power. In usually the operation of excess power uses power curtailment by the solar inverter and reveals a waste of excess PV power. Therefore, the excess electricity fraction is limited to 3% to a novel precise analysis and effective use of energy for the economic benefit evaluation of PV/BESS. The energy arbitrage revenue can be obtained by calculating the optimal economical capacity allocation of a PV system and BESS in the conditions of meet load demand and PV’s RF constraints at the same time.

Please refer to line 139 to 151.

Point 7: Writing could be improved.

Response 7:

After checking, the English writing and grammar errors have been checked and fixed by a commercial service in grammarly.com.

Round 2

Reviewer 1 Report

Responses to question 1 : you wrote more information on the objectif of the paper but not on the originality compared with previous works. I continue to say that this pape ris an application of an existing software and is not a research paper.

The unities were not changed as asked during the first review: the numbers you used have too much digits

Figure 5 : there is a problem, the peak power is not 7,378kW as said in the text.

Response 5: no you don’t answer to my question: you will discharge the battery according to the load to satisfy and not with a constant C-rate. Your answer is not satisfactory

Response 6: not satisfactory. You don’t explain the calculation of the replacement cost,

Response 10: no satisfactory

We consider that the paper was not improved.

Author Response

Dear Reviewer,

According to reviewer’s comments the major revisions had been completed using the “the parts marked in red with any change”. The cover letter to each reviewer provides the explanations, addressing certain comments with point by point responses to the referee’s comments as possible as we can. We greatly appreciate the comments of reviewer on our manuscript. These changes have clearly improved our manuscript.

Best wishes,

Cheng-Chien Kuo

Reply to reviewer’s comments:

Point 1: Responses to question 1 : you wrote more information on the objectif of the paper but not on the originality compared with previous works. I continue to say that this pape is an application of an existing software and is not a research paper.

Response 1:

Many thanks for the reviewer’s commend.

The HOMER software is a simulation analysis tool. It is not easy to directly produce the required results with the software. The analysis method requires users to plan according to their needs, and can carry out relevant research and analysis according to different research needs. This study uses HOMER simulation to propose the economic benefit analysis method of PV/BESS applied to BTM, which is to consider the effective use of renewable energy. Especially an analysis method is considered by the restriction of excess electricity, and to filter out the solutions that meet the restriction of excess electricity from the simulation results (excess electricity is set to 3% in this study) to find out the capacity of PV/BESS solutions under the conditions of different proportions of renewable energy. The analysis method to most project benefit is the part that has not been considered in the economic benefit analysis literature in the past, and this method is proposed for the engineer application of renewable energy to the BTM economic benefit analysis. The design plan is under comprehensive considerations for the electricity users to evaluate and choose the system plan.

Please refer to the introduction part.

Point 2: The unities were not changed as asked during the first review: the numbers you used have too much digits

Response 2:

Many thanks for the reviewer’s commend.

We modify the units and the numbers expressed in scientific notation to be consistent throughout the paper.

Please refer to the number parts marked in red in the article.

Point 3: Figure 5 : there is a problem, the peak power is not 7,378kW as said in the text.

Response 3:

Many thanks for the reviewer’s commend.

Figure 5 shows the load profile based on the power data of users in 2019. The power data imported into the software raw data is one data every 5 minutes. The load power in the 24 hours in Figure 5 is the average value of 365 days in each period. The peak power of 7,378kW is the highest load power ever experienced in the year.

Please refer to line 263 to 267.

Point 4: Response 5: no you don’t answer to my question: you will discharge the battery according to the load to satisfy and not with a constant C-rate. Your answer is not satisfactory

Response 4:

Many thanks for the reviewer’s commend.

In this study, the high-energy-density and high-safety lithium ferrous phosphate battery (LFP) was used for economic benefit analysis. The battery cell used in this study has a rated specification of 3.2V/280Ah. The upper limit of the current can allow for continuous charging or discharging with 140A, and the time would require to fully charge or discharge with 2 hours, which is a 0.5 C-rate battery. The HOMER simulation software adjusts the current according to the power demand for charging or discharging, and the upper limit is 140A, which is not a constant value. The battery module is made up of 14 battery cells in series with a rated power of 44.8V/280Ah, and the battery rack is made up of 17 battery modules in series with a rate of 761.6V/280Ah and a rated capacity of 213.2kWh. The operating voltage of the battery cabinet is 666.4~856.8V, the minimum state of charge (SOC) is set to 20%, and the round trip efficiency is set to 92%.

Please refer to line 450 to 460.

Point 5: Response 6: not satisfactory. You don’t explain the calculation of the replacement cost,

Response 5:

Many thanks for the reviewer’s commend.

The lifetime presents the number of discharge-charge cycles to meet specific performance criteria of the battery. The operating lifetime throughout of 967.765MWh of the battery is affected by the parameters (the rate and depth of cycles) and by the environmental conditions (temperature and humidity). The warranty of a commercial battery is the ranges from 5-10 years according to industrial experiences. The related standard such as IEC62620 can provide the parameters to lifetime according to the customer’s declarations. The battery wear cost is the cost of cycling energy through the storage system calculated by Equation (13), and the storage life is limited by throughput. HOMER assumes the storage requires replacement when its total throughput equals its lifetime throughput, and the battery storage approaches its required replacement. HOMER calculates the storage wear cost using the following equation:

                                                                                                                (13)

Cbw = the battery wear cost ($)

Crep,batt = the replacement cost of the storage ($)

Nbatt = the number of batteries in the storage

Qlifetime = the operating lifetime throughput of a single storage (kWh)

ηrt = storage roundtrip efficiency (fractional)

Please refer to line 461 to 477.

Point 6: Response 10: no satisfactory

We consider that the paper was not improved.

Response 6:

Many thanks for the reviewer’s commend.

We have revised for a decimal point and contract capacity unit in MW in the manuscript. We also modify the units and the numbers expressed in scientific notation to consistent throughout the paper.

Please refer to the number parts marked in red in the article.

Reviewer 2 Report

The manuscript can be published as-is.

Author Response

Dear Reviewer,

We greatly appreciate your comments on our manuscript. These changes have clearly improved our manuscript. Many thanks for the reviewer.

Best wishes,

Cheng-Chien Kuo

Reviewer 3 Report

Please write the organisation of the work at the end of the Introduction Section. Each reference should be described with a description. Please do not use a lump sum reference.

Author Response

Dear Reviewer,

According to reviewer’s comments the revisions had been completed using the “the parts marked in red with any change”. The cover letter to the reviewer provides the explanations, addressing certain comments with point by point responses to the referee’s comments as possible as we can. We greatly appreciate the comments of reviewer on our manuscript. These changes have clearly improved our manuscript.

Best wishes

Cheng-Chien Kuo

Reply to reviewer’s comments:

Point 1: Please write the organisation of the work at the end of the Introduction Section. Each reference should be described with a description. Please do not use a lump sum reference.

Response 1:

For this comment, we have improved and revised the manuscript for each reference description using the “the parts marked in red with any change”. We add the organization of the work at the end of the Introduction Section.

This paper uses this tool to propose the economic benefit analysis method of PV/BESS applied to BTM, which is to consider the effective use of renewable energy. Especially an analysis method is considered by the restriction of excess electricity, and to filter out the solutions that meet the restriction of excess electricity from the simulation results (excess electricity is set to 3% in this article) in order to find out the capacity of PV/BESS solutions under the conditions of different proportions of renewable energy. The analysis method to most project benefit is the part that has not been considered in the economic benefit analysis literature in the past, and this method is proposed for the engineer application of renewable energy to the BTM economic benefit analysis. The design plan is uner comprehensive considerations for the electricity users to evaluate and choose the system plan.

Please refer to the introduction section, line 165 to 175

Round 3

Reviewer 1 Report

I am sorry to continue to say that this paper is an application of an existing software and not a research paper. I thought that this journal published research papers, there is no novelties in the method and the authors always answer that the HOMER software does this thing or this thing.

Line 161: you are researchers thus using HOMER must not be a difficult task; how can you tell than a commercial software is difficult to use. I am not against this software but I say that it is useful for engineering calculation but cannot be used for a research work.

Introduction: the authors wrote nothing on the novelty of this work.

Author Response

Dear Reviewer,

According to reviewer’s comments, the revisions had been completed using the “the parts marked in red with any change”. The cover letter to the reviewer provides the explanations, addressing certain comments with point by point responses to the referee’s comments as possible as we can. We greatly appreciate the comments of reviewer on our manuscript. These changes have clearly improved our manuscript.

Best wishes,

Cheng-Chien Kuo

Reply to reviewer’s comments:

Point 1: I am sorry to continue to say that this paper is an application of an existing software and not a research paper. I thought that this journal published research papers, there is no novelties in the method and the authors always answer that the HOMER software does this thing or this thing.

Line 161: you are researchers thus using HOMER must not be a difficult task; how can you tell than a commercial software is difficult to use. I am not against this software but I say that it is useful for engineering calculation but cannot be used for a research work.

Introduction: the authors wrote nothing on the novelty of this work.

Response 1:

Many thanks for the reviewer’s commends.

(1) Our previous explanation may make you misunderstood, we have modified and revised the descriptions of HOMER software to readers understand more clearly.

For using HOMER software to execute the projects, there are many parameters that need to be approximately applied according to their purpose and practical experience with domain knowledge to plan, modification and operation. Although this paper is not theoretical research, it is a practical site for establishing the systems consisted of PV/BESS through HOMER’s operation and planning arrangements.

(2) We add 6 references in Table 1 to list the hybrid energy systems with various project scenarios.

(3) The descriptions focus on their application purposes and practical experiences.

The issue of access to sustainable energy sources is crucial to support the healthcare facilities to deliver services under a grid-connected or an off-grid RHU (rural health units). Based on the experience, the identified load profiles of equipment can be generally grouped as medical equipment, HVAC (heating, ventilation, and air conditioning), lighting, and office equipment used throughout the day, specifically for the common appliances used in an RHU [29]. In order to mitigate the dependency on diesel generators where grid extension is not feasible, the generation of electricity may be effectively utilized to the applications of isolated inhabitants through stand-alone PV power systems. The techno-economic analysis is performed to obtained optimal size by empirically assuming typical seasonal load profiles of three distinct island seasons for a single household [30]. To alleviate this challenge of the intermittent nature of renewable energy sources (RESs), it is common practice to integrate RESs by efficient batteries playing the leading role utilized as stationary energy storage systems. The investigations can be concluded that Li-ion batteries require 40% lesser batteries as compared to lead-acid batteries, and Li-ion batteries provided lower NPC and COE for photovoltaic grid-connected systems [31]. Due to the low price of diesel fuel in Iran, the competitiveness of such renewable systems with diesel generator systems is low. To increase the economic acceptability, the analysis leads to that Iran requires the government’s support and incentive schemes to the competitiveness in energy supply systems [32]. The electrification of consumption is implemented for three different PV residential households with incorporating feed-in Tariff of day (ToD) tariffs regulation/ net metering process. The feasibility presents the executable option and the possible consequence of social benevolent policy for the dissemination of decentralized grid-connected renewable energy systems [33]. Under the local policies’ limits on maximum PV size in Queensland, the maximized PV size is able to maximize the Queensland residents’ benefits and meets the requirement of minimizing the total costs of system investment related to electricity consumption during the system’s lifetime. The result was also obtained that the best slope of the PV panel located at 20o~ 25o dependent on the different cities [34].

Please refer to line 130~163.
